

# A Global Database of Marine Isotope Stage 5a and 5c Marine Terraces and Paleoshoreline Indicators

Schmitty B. Thompson[1] and Jessica R. Creveling[1]

[1]College of Earth, Ocean, and Atmospheric Sciences, Oregon State University, Corvallis, OR, USA, 97331

*Correspondence to*: Schmitty B. Thompson (thomschm@oregonstate.edu)

**Abstract.** In this review we compile and detail the elevation, indicative meaning, and chronology of Marine Isotope Stage 5a and 5c sea level indicators for 39 sites within three geographic regions: the Pacific coast of North America, the Atlantic coast of North America and the Caribbean, and the remaining globe. These relative sea level indicators, comprised of geomorphic

indicators such as marine and coral reef terraces, eolianites, and sedimentary marine and terrestrial limiting facies, facilitate future investigation into Marine Isotope Stage 5a and 5c interstadial paleo-sea level reconstruction, glacial isostatic adjustment, and Quaternary tectonic deformation. The open access database, presented in the format of the World Atlas of Last Interglacial Shorelines (WALIS) database, can be found at https://doi.org/10.5281/zenodo.4426206 (Thompson and Creveling, 2021).

## 1 Introduction

Two orbitally modulated peaks in northern hemisphere summer insolation, occurring ~100 and ~80 ka, brought warmer temperatures and reduced ice volumes that briefly interrupted earth's transition from the last interglacial into the last glacial maximum (Milankovitch *et al.*, 1938; Hays *et al.*, 1976; Chappell and Shackleton, 1986; Lambeck and Chappell, 2001; Cutler *et al.*, 2003). These interstadials induced $\delta^{18}O_{benthic}$ excursions, designated as Marine Isotope Stages (MIS) 5c and 5a,

coincident with high stands in sea level inferred from uplifted reef and wave-cut terraces and additional sedimentological indicators (Mesolella *et al.*, 1969; Railsback *et al.*, 2015). Inquiry into MIS 5a and 5c sea level high stands enriches our understanding of last interglaciation (*sensu lato*) paleoclimate (e.g., Potter et al., 2004) and tectonic deformation (e.g., Simms et al., 2016), and faunal assemblages preserved on and within marine terraces reveal ocean paleo-temperature and paleo-circulation pathways (Muhs *et al.*, 2012), all of which complement insight gained from the preceding MIS 5e substage

(Kopp et al., 2009; Dutton and Lambeck, 2012; Dutton et al., 2015).

A rich literature catalogues the legacy of mapping globally distributed MIS 5a and 5c reef tracts, wave-cut terraces, and other marine- and terrestrial-limiting sedimentological indicators for the purpose of measuring the local peak sea level achieved during these ice-volume minima (Griggs, 1945; Alexander, 1953; Bretz, 1960; Land *et al.*, 1967; Mesolella, 1967; Chappell,





1974; Chappell and Veeh, 1978; Cronin *et al.*, 1981). Here we adopt the standardized framework provided by the World Atlas of Last Interglacial Shorelines (WALIS) database (WALIS, https://warmcoasts.eu/world-atlas.html) to compile the English-language publications of globally outcropping relative sea level (RSL) indicators ascribed by the primary authors as MIS 5a and 5c in age. The open-access database, which includes site descriptions, elevation and geochronological constraints, and associated metadata, is available at this link: https://doi.org/10.5281/zenodo.4426206 (Thompson and

Creveling, 2021). Database field descriptors can be queried at this link: https://doi.org/10.5281/zenodo.3961543 (Rovere et al., 2020). This database builds on foundational regional syntheses of MIS 5a and 5c sea level indicators for the Atlantic coast of North America and the Caribbean (Potter and Lambeck, 2003), Pacific coast of the United States and Baja California, Mexico (Muhs et al., 2012; Simms et al., 2016), and a subset of far-field localities (Creveling et al., 2017) in compiling a global dataset of 36 sites with MIS 5a and 5c paleo-sea level indicators (Figure 1). The database includes site

excluded from previous reviews. The following sections include a summary of the types of geomorphic and sedimentological sea level indicators included in this review (Section 2); details on how elevation measurements, uncertainty, and sea level data are reported (Section 3); and an overview of the dating methods utilized in the primary publications (Section 4). The majority of this publication (Section 5) reports the current measured elevations and chronologies, along with the history of the literature, for individual sites. Section 6 summarizes the content of this review for

future research directions. Section 7 addresses the data availability.

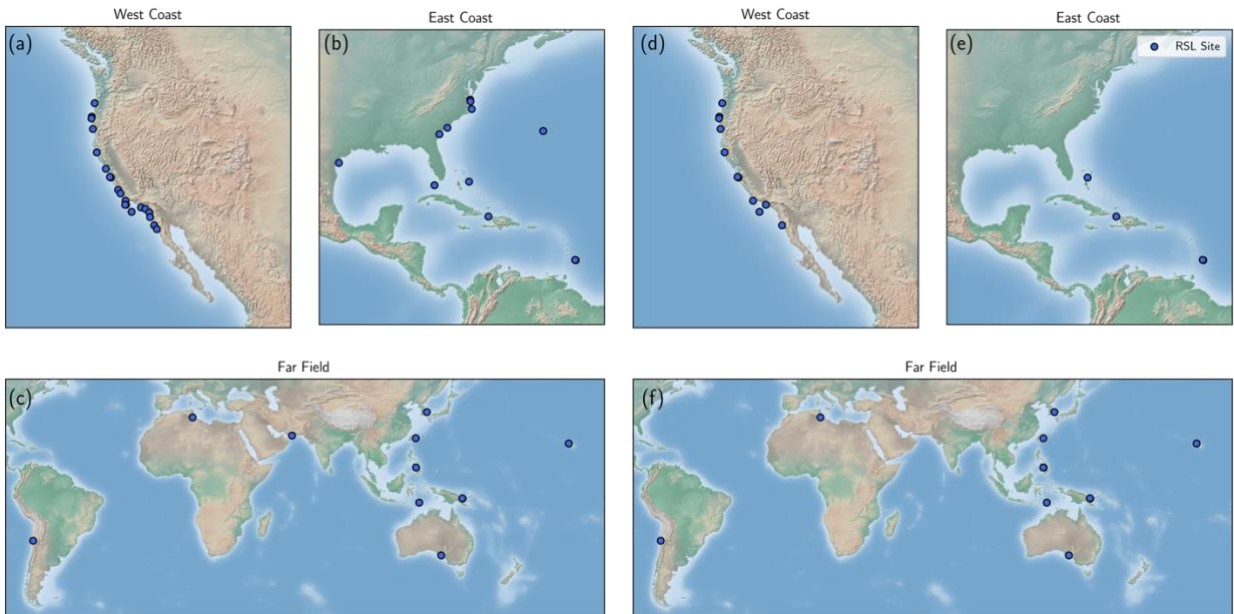

Figure 1: Locations of MIS 5a (a–c) and MIS 5c (d–e) relative sea level (RSL) indicator sites for (a,d) the Pacific coast of North America, (b,e) Atlantic coast of North America and the Caribbean, and (c,f) the remaining globe atop the Matplotlib Basemap Shaded Relief map (Hunter, 2007).


The present elevation of MIS 5a and 5c sea level indicators reflect a number of convolved processes, including, but not limited to, tectonic deformation and glacial isostatic adjustment, which require carefully applied corrections to reconstruct peak global mean sea level (GMSL). Tectonic deformation can alter the elevation of an indicator by tens to hundreds of meters (Alexander, 1953; Chappell, 1974), and glacial isostatic adjustment by similar magnitudes (Creveling *et al*., 2015; Simms *et al*., 2016). For active margins, this convolution serves as an opportunity to constrain rates of Quaternary tectonic deformation (Adams, 1984; Marquardt *et al*., 2004; Muhs *et al*., 2014) given robust assumptions of the magnitude and source of MIS 5e ice-volume melt and glacial isostatic adjustment (Broecker *et al*., 1968; Dodge *et al*., 1983; Chappell and Shackleton, 1986; Creveling et al., 2015; Simms *et al*., 2016). Similarly after correction for tectonic uplift, discrepancies in the local elevation of globally distributed MIS 5a and 5c indicators of peak sea level retain a meaningful signal of glacial isostatic adjustment (GIA; Potter and Lambeck, 2003) which, in turn, allows for the reconstruction of peak global mean sea level (GMSL) and assessments of the sensitivity of Quaternary ice sheets to the influence of Milankovitch forcing on climate in sub-100 kyr timescales (Lambeck and Chappell, 2001; Potter and Lambeck, 2003; Potter *et al.,* 2004; Muhs *et al*., 2012; Simms *et al*., 2016; Creveling *et al*., 2017). We emphasize that this database reports measurements of uncorrected, present-day elevation of various relative sea level indicators such that will enable the user to apply corrections based on the most current data- and model-based predictions.

## 2 Sea Level Indicators

The data detailed in this review (and the associated database) comprises a set of geomorphic and sedimentological indicators of past sea level, for which a comprehensive overview can be found in Rovere *et al.* (2016). Wave-cut bedrock marine terraces make up the largest portion of these data, particularly for the west coast of North America (see Muhs et al., 1992b for a detailed description of this indicator). Constructional coral reef terraces are prevalent across the east coast of North America, the Caribbean, and the far field regions (see Chappell (1974) for an example of this indicator). Additional sedimentary features, such as eolianite, submerged beach ridges, and exposure surfaces, make up the remainder of the local inferences.

## 3 Elevation Measurements

Here we catalogue the elevation, with uncertainty (if listed), of a given indicator as reported in the primary publication(s) without modification. Methods adopted to measure the present-day elevation range of reported indicators vary from hand level and altimeter surveys to mapping with modern differential GPS and Digital Elevation Models. The majority of publications summarized herein did not report the sea level datum used and, thus, these methods are reported in the WALIS database as "General Definition" as defined in the metadata. For those sites for which primarily field workers did not report elevation uncertainty, we noted this absence in the database and assigned a measurement uncertainty based upon the defined



accuracy of the elevation measurement method. When available in the original publication, latitude and longitude coordinates for indicator sites are reproduced for the WALIS database; when unavailable, coordinates were interpreted from publication maps using Google Earth and noted accordingly.

## 4 Dating Techniques

Age assignments for MIS 5a and 5c sea level indicators arise from a wide variety of radiometric and non-traditional geochronologic methods. Absolute chronologies for indicators composed of *in situ* carbonate utilize uranium-thorium dating (Barnes *et al*., 1956; Broecker and Thurber, 1965; Osmond *et al*., 1965; Thurber *et al*., 1965; Muhs *et al*., 2002; 2012). Sediment mantling sea level indicators can also yield absolute chronologies through luminescence dating (Duller, 2004; Grove *et al*, 2010). Amino acid racemization creates relative chronologies, especially in conjunction with other amino acid

ratios from sea level indicators benchmarked by radiometric dates (Mitterer, 1974; Miller *et al*., 1979; Kennedy *et al.,* 1982). Stratigraphic relationships between adjacent sea level indicators have been extensively used to develop relative chronologies for sea level indicators, especially for sites with marine terrace sequences consisting of adjacent MIS 5e, 5c, and 5a terraces (Adams *et al*., 1984; Merritts and Bull, 1989; McInnely and Kelsey, 1990). Other sparingly used dating methods for MIS 5a and 5c indicators include: electron spin resonance (Mirecki *et al*., 1995), terrestrial cosmogenic nuclide dating (Perg *et al*.,

2001), Protactinium-231 dating, (Edwards *et al*.,1997), paleomagnetic stratigraphy (see sources discussed in Choi *et al*., 2008), soil development stages (Kelsey *et al*., 1996), radiocarbon (Hanson *et al*., 1992), and geomorphic models (Hanks *et al*., 1984; Valensise and Ward, 1991).

## 5 A Global Database of Marine Isotope Stage 5a and 5c Relative Sea Level Indicators

### 5.1 West Coast of North America

Field observers first documented emergent marine terraces on the west coast of North America in the early 20th century. Early mapping efforts documented terraces along much of the California coastline (Ellis 1919; Davis, 1932; Woodring *et al.* 1946). Alexander (1953) pioneered the interpretation of west coast marine terraces as indicators of paleo-sea level and tectonic uplift. Since then, extensive documentation of regional MIS 5a and 5c paleo sea level indicators has yielded 18 sites

which are included in the following review. As noted below, many such studies reported geomorphic or chronological data that applies to multiple adjacent sites. Griggs (1945) completed early work on the Oregon coast, documenting four terraces, for which we focus on the Whiskey Run and Pioneer terraces. McInelly and Kelsey (1990) provided further geomorphic cross sections for the Whiskey Run and Pioneer terraces at the Cape Arago and Coquille point sites. Early chronologies for many west coast sites come from Kennedy *et al.* (1982), which provided Leucine D:L ratios for individual sites plotted

against isochrons independently constrained by uranium-series dates. These sites were summarized by Simms *et al*. (2016), which compared tectonically corrected RSL sites to models of west coast glacial isostatic adjustment.

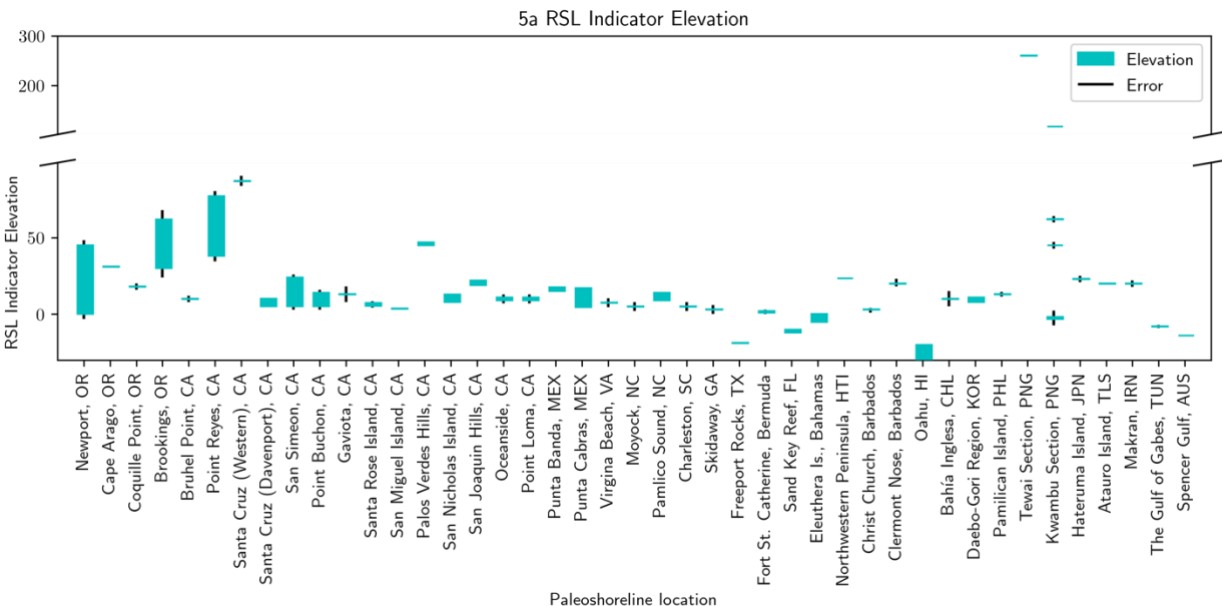

**Figure 2: The present elevation of MIS 5a relative sea level (RSL) indicators, in meters, at the field locations listed on the horizontal axis. Teal bars represent the full elevation range reported in the primary publications along with the corresponding**
**measurement error, if any, in black bars. Note the scale break on the vertical axis necessary to present indicator elevations for sites with rapid tectonic uplift.**

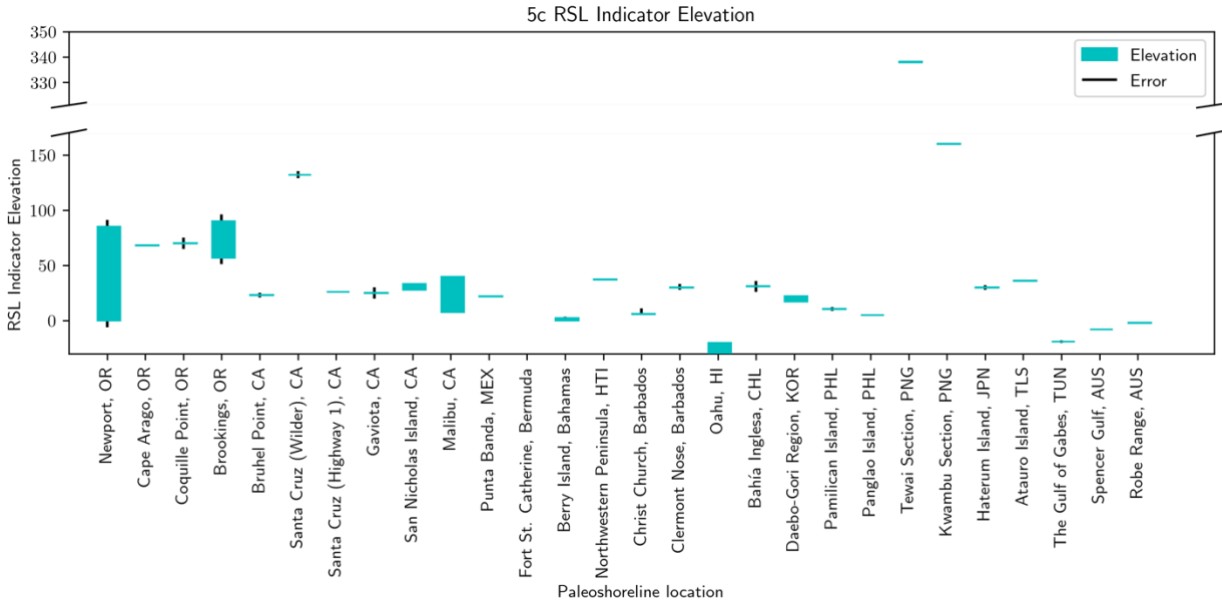

**Figure 3: The present elevation of MIS 5c relative sea level (RSL) indicators, in meters, at the field locations listed on the horizontal axis. Teal bars represent the full elevation range reported in the primary publications along with the corresponding**

**measurement error, if any, in black bars. Note the scale break on the vertical axis necessary to present indicator elevations for sites with rapid tectonic uplift.**



**Figure 4: Age assignments for MIS 5a relative sea level (RSL) indicators cropping out at the locations listed on the horizontal axis. For each location, the chronology method is indicated by color, absolute/minimum/maximum age type by shape, and the confidence of the date by border color. Symbols represent absolute numerical dates whereas, in the absence of an absolute chronology, bars represent general Marine Isotope Substage assignments (MIS 5c 98 – 90 ka; MIS 5a 86 – 72 ka).**



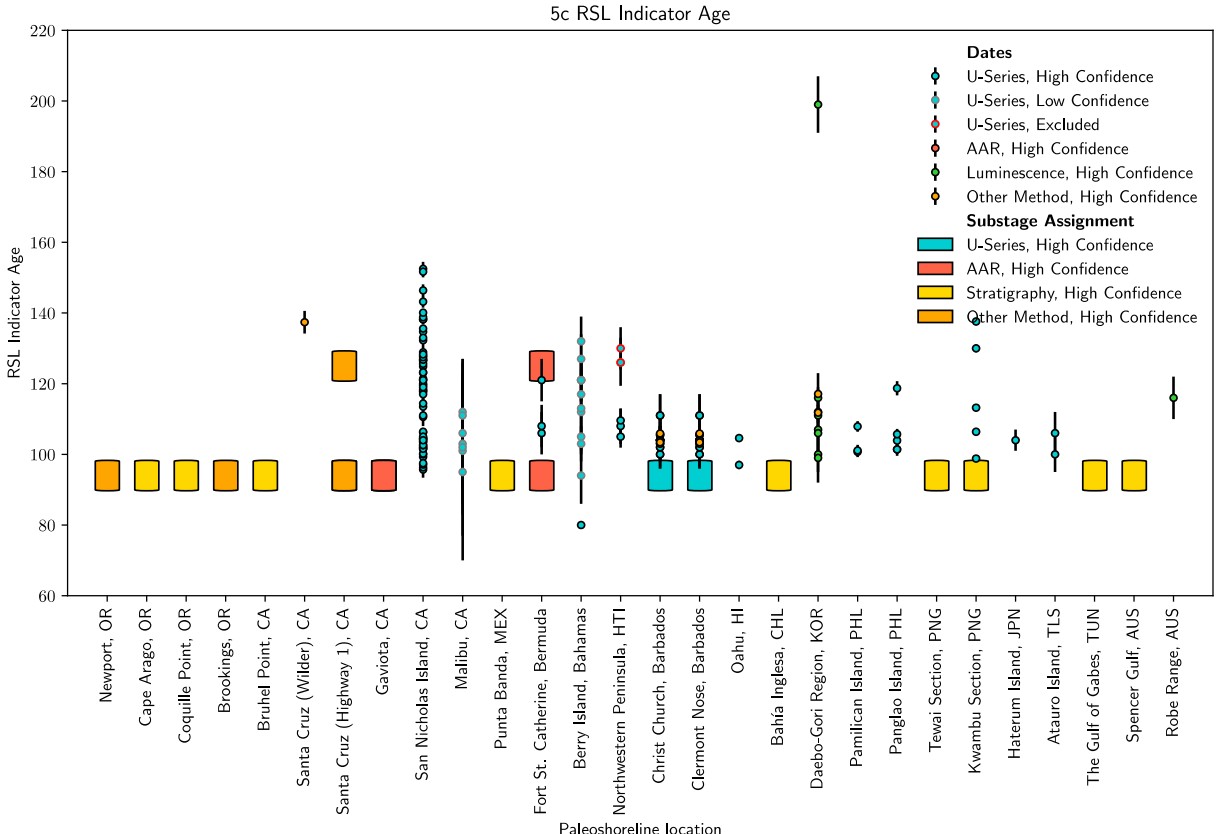

**Figure 5: Age assignments for MIS 5c relative sea level (RSL) indicators cropping out at the locations listed on the horizontal axis. See Figure 4 caption for a description of symbols.**


### 5.1.1 Newport, Oregon

Kelsey *et al.* (1996) utilized topographic maps and altimeter surveys to map and document the platform elevation ranges of six emergent bedrock terraces which crop out discontinuously through the region due to faulting; we focus on the lowest two of the surveyed platforms, the Newport and Wakonda terraces. Kennedy *et al.* (1982) first dated the Newport terrace using

amino acid racemization (AAR). The Leucine D:L ratio was plotted against isochrons of other AAR dates from proximal locations, assigning the Newport terrace to MIS substage 5a. Kelsey *et al.* (1996) assigned terraces to MIS substages based on soil development stages. Namely, this study assigned the Newport terrace (0 – 45 m apsl; above present day sea level), which is extensive north of Yaquina Bay and discontinuous between Cape Foulweather and Siletz Bay, to MIS 5a; the Wakonda terrace (0 – 85 m apsl)—which is discontinuous from Newport north to Otter Rock, and then crops out just above

modern beach elevation between Yaquina and Alsea bays before gradually descending below sea level south of Alsea Bay— was assigned to MIS 5c. Based on these age assignments, Kelsey *et al.* (1996) correlated the Newport and Wakonda terraces

to the Whiskey Run and Pioneer terraces (see Cape Arago below). Figures 2 and 3 include the terrace elevations reported by Kelsey *et al.* (1996) and Figures 4 and 5 illustrate the MIS substage assignments for the Newport and Wakonda terraces, respectively (Kennedy *et al.*, 1982; Kelsey *et al.*, 1996).

**5.1.2 Cape Arago, Oregon**

Griggs (1945) mapped four emergent wave-cut terraces along the central Oregon coastline for which we focus on the lowest two, the Pioneer and Whisky Run terraces. Adams (1984) documented landward tilting of the Pioneer and Whisky Run terraces and the faults that vertically displace them. No radiometric dates directly constrain the age of the Pioneer and Whisky run terraces at Cape Arago; instead, these terraces are assigned to MIS 5a and MIS 5c based on correlation to

terraces cropping out at Coquille Point (see below; Figures 4 and 5). McInelly and Kelsey (1990) published altimeter surveys that revised the peak shoreline angle elevations of the Whisky Run and Pioneer terraces reported by Adams (1984) to 31 m apsl and 68 m apsl, respectively (Figures 2 and 3). McInelly and Kelsey (1990) also reported the thickness of the sediment packages overlying both terraces and revised the mapped faults that displace each terrace.

**5.1.3 Coquille Point, Oregon**

Griggs (1945) mapped four emergent, wave-cut terraces along the central Oregon coastline for which we focus on the lowest two, the Pioneer and Whisky Run terraces. Attribution of the Whisky Run terrace to the MIS 5a substage first appeared in Kennedy *et al.* (1982) who reported Leucine D:L ratios on *Saxidomus* and a single uranium-series date on a coral (Fig. 4). Subsequently, six uranium-series dates on corals and bryozoans, 10 amino acid ratios on bivalve mollusks *Mya truncata* and *Saxidomus giganteus*, and oxygen isotope stratigraphy on mollusk shells supported the MIS 5a age assignment for Whiskey

Run terrace (Kennedy *et al.*, 1982; Muhs *et al.*, 1990; Muhs *et al.,* 2006; Fig. 4). McInelly and Kelsey (1990) revisited the region and presented a representative geomorphic cross-section that documented deformation of the Whisky Run terrace by the Pioneer anticline. This survey reported a Whisky Run terrace maximum elevation of 18 m apsl (Figure 2), a small upward revision from the 17 m apsl reported by Muhs *et al.* (1990). No elevation for the Pioneer terrace was reported in the text of this article, though Simms *et al.* (2016) extracted an elevation of 70 m apsl from the cross-section of McInelly and

Kelsey (1990). Based on their similar elevations, McInelly and Kelsey (1990) correlated the Pioneer terrace at Coquille Point to the MIS 5c Pioneer terrace at Cape Blanco, itself dated through amino acid ratios and faunal assemblages (Muhs *et al.,* 1990; Figure 5).

**5.1.4 Brookings, Oregon**

Kelsey and Bockheim (1994) utilized topographic maps to document seven wave-cut terraces; the lowest two terraces, Harris

Butte and Brookings, crop out at bedrock platform elevations of 30 – 62 m apsl and 57 – 90 m apsl (see Figures 2 and 3), respectively, south of the Whaleshead fault zone. No radiometric dates constrain the age of the Harris Butte and Brookings

terraces. These terraces were assigned to MIS 5a and 5c, respectively, based upon similar soil development stages to the Whisky Run and Pioneer terraces at Cape Arago (Figures 4 and 5).

### 5.1.5 Bruhel Point, California

Merritts and Bull (1989) surveyed the 14 marine terraces cropping out at Bruhel Point, for which the 10 m apsl and 23 m apsl terraces (surveyed from the inner edge of the terrace) have been assigned to the MIS 5a and 5c high stands (Figures 2 and 3). A Leucine D:L ratio on the 10 m apsl terrace indicates either a MIS 5a or 5c stage designation (Kennedy *et al.*, 1982). In contrast, Merritts and Bull (1989) assigned the 10 m apsl and 23 m apsl terraces to MIS 5a and 5c, respectively (see Figures 4 and 5), based upon correlations made using diagrams of inferred uplift of the Bruhel Point terraces vs inferred

ages (uplift-rate diagrams) to the New Guinea sea level curve (see Tewai and Kwambu sections).

### 5.1.6 Point Reyes, California

Grove *et al.* (2010) utilized differential GPS to map nine marine terraces across five transects for which the inner edge of the lowest terrace, of purported MIS 5a, was surveyed between 38–77 m apsl (Figure 2). Five sediment samples overlying the lowest terrace yielded three dates per sample from three energy stimuli: optically stimulated blue-light luminescence (OSL),

optically stimulated infra-red luminescence (IRSL), and thermoluminescence (TL). Grove *et al.* (2010) primarily discounted the OSL dates for being too young (MIS 2–3) whereas the TL dates were interpreted as maximum ages; the IRSL dates were identified as the most accurate estimates of the time of deposition. The sample identified as the best indicator of terrace age (PR-2) assigns the lowest terrace to MIS 5a; the other four samples yield ages ranging from MIS 3 to MIS 5a or show internal inconsistencies (Figure 4).

### 5.1.7 Santa Cruz, California

Alexander (1953) first mapped the emergent marine terraces cropping out in the Santa Cruz, CA region, for which we focus on (often conflicting) interpretations for the age of the laterally extensive, sediment mantled terrace formerly named the Santa Cruz terrace, as well as the two higher elevation Western and Wilder terraces later documented by Bradley and Griggs (1976). Bradley and Addicott (1968) presented four uranium-series dates on mollusks that assigned an age to the Santa Cruz

terrace—referred to as the "first" terrace in their nomenclature—consistent with either MIS 5a or 5c. Bradley and Griggs (1976) utilized seismic surveys to subdivide the Santa Cruz terrace into three constituent terraces which, in ascending elevation order include: the Davenport, Highway 1, and Greyhound terraces. A rich literature discusses the chronostratigraphic assignment of the Davenport and Highway 1 terraces. Most authors have concluded, on the basis of amino acid ratios, diffusion modeling of paleo-sea cliffs, and 15 U-series dates on corals that the Davenport terrace, which

crops out at 5 - 10 m apsl (see Muhs *et al.*, 2006), formed during MIS 5a (Kennedy *et al.*, 1982; Hanks *et al.*, 1984; Muhs *et al.*, 2006). The Highway 1 terrace, at an inner edge elevation of ~26 m apsl, has been alternately assigned to MIS 5c using models of the diffusion of paleo-sea cliffs (Hanks *et al.*, 1984) or to MIS 5e based upon a geologic fault modeling (Valensise



and Ward 1991). In contrast to the above chronology, Perg *et al.* (2001) presented 10 cosmogenic nuclide dates that assigned the Santa Cruz (undifferentiated), Western (87 m apsl; Figure 2), and Wilder (132 m apsl; Figure 3) terraces to MIS 3, 5a, and 5c respectively. Muhs *et al.* (2006) contested these cosmogenic dates, arguing that they date the deposition of alluvium overlying the terraces and therefore provide a minimum age for terrace formation. We report Davenport/Highway 1 and Western/Wilder terrace elevations and chronologies in the database as separate entries in the database (see Figures 2 – 5) to most accurately represent the existing body of literature. For each terrace all of the reported dates and age assignments are represented, even when conflicting.

### 5.1.8 San Simeon, California

Hanson *et al.* (1992) mapped five wave-cut terraces along the San Simeon fault zone. North and south of the fault zone the San Simeon terrace shoreline angle crops out discontinuously between 5 – 7 m apsl and 9 – 24 m apsl, respectively (Figure 2). While a uranium-series dates on bone and a radiocarbon date on detrital charcoal provide a minimum terrace age of MIS 3 (Hanson *et al*., 1992), two thermoluminescence dates on sediment underlying the emergent platform place the San Simeon terrace within the range of MIS 5a to 5c (Berger and Hanson, 1992). Of these possibilities, Simms *et al.* (2016) elected for the MIS 5a San Simeon terrace age assignment (Figure 4).

### 5.1.9 Point Buchon, California

Hanson *et al.* (1992) mapped a flight of marine terraces across south-central California; the lowest terrace, Q1, with surveyed shoreline angles between 5 – 14 m apsl, is overlain by up to 2 m of marine sediment and 15 – 30 m of non–marine sediment and is cut by several reverse faults (Figure 2). The Q1 terrace was assigned to MIS 5a based upon three minimum ages from uranium-series dates on marine and terrestrial mammal teeth and bone found in the overlying sediment deposits (Figure 4) Two uranium series dates on coral are included which were noted to be unreliable by the authors.

### 5.1.10 Gaviota, California

Rockwell et al. (1992) mapped five well-expressed marine terraces using transit-stadia surveys; the lowest terrace, the Cojo terrace (10 – 17 m apsl shoreline angle; Figure 2), is found west of the South Branch Santa Ynez fault (SBSYF) and crosses the hinge of the Government Point Syncline. Seven uranium-series dates on bone and mollusk, two of which are maximum ages, eight amino acid ratios, and the cool-water aspect of the terrace fauna assign the Cojo terrace to MIS 5a (Rockwell et al., 1992; Kennedy et al., 1992; Figure 4). Rockwell et al. (1992) correlated the first emergent terrace east of the SBSYF to the Cojo terrace based upon similar amino acid ratios and terrace faunal aspect. While the elevation of the overlying second terrace at this locale was not reported by Rockwell et al. (1992), Simms et al. (2016) extracted an elevation of 25 m apsl from an illustration of the shore-parallel terrace profile (see Fig. 2 of Rockwell et al., 1992; Figure 3). No radiometric dates exist for the second terrace; Rockwell et al. (1992) inferred an MIS 5c age for the second terrace using stratigraphic correlation to the 5a terrace and the cool–water aspect of the terrace fauna (Figure 5).



### 5.1.11 California Channel Islands (San Miguel and Santa Rosa islands)

Orr (1960) first mapped marine terraces on northwestern Santa Rosa Island and later efforts by Dibble and Ehrenspeck (1998) and Pinter *et al*. (2001) elaborated on both terrace mapping and the geology of the island. Muhs *et al*. (2014) utilized differential GPS to map the flight of emergent marine terraces on San Miguel and Santa Rosa Island, for which we focus on the lowest terrace at Santa Rosa Island, with a shoreline angle mapped between 5.4 – 7.4 m apsl, and the lowest terrace at San Miguel Island, with a shoreline angle mapped at ~3.5 m apsl (Figure 2). Two amino acid ratios on *Chlorostoma* shells

assigned the lowest Santa Rosa Terrace terrace to MIS 5a (Figure 4). The lowest San Miguel terrace is overlain by a veneer of fossiliferous. cemented gravel and marine sand, which itself is overlain in areas by alluvium (Johnson, 1969; Muhs *et al.*, 2014). Based on stratigraphic position, Muhs *et al.* (2014) assigned the San Miguel terrace to MIS 5a (Figure 4).

### 5.1.12 Palos Verdes Hills, California

Woodring *et al*. (1946) conducted early mapping of the 13 emergent, wave-cut terraces of the Palos Verdes Peninsula; seven

early uranium-series dates, each corrected for open-system histories, assigned the first (lowest) terrace to MIS 5a (Szabo and Rosholt 1969). Muhs *et al*. (1992a) showed, on the basis of aminostratigraphy on *Tegula* and *Protothaca*, and oxygen isotope data on *Epilucina,* that the 'first terrace', as mapped by Woodring *et al*. (1946), instead represents high stand deposits of both MIS 5a and 5e. Muhs *et al*. (2006) refined map units and assigned place-based names to supplant the counting scheme of Woodring *et al*. (1946), redefining the lowest, horizontally continuous surface as the Paseo del Mar

terrace (the 'second' terrace of Woodring *et al*., 1946) with an estimated shoreline elevation angle of ~45 – 47 m apsl (Figure 2). A combination of 13 uranium-series dates on *Balanophyllia,* and extralimital northern species within the faunal assemblage, assigns the Paseo del Mar terrace to MIS 5a (Figure 4). Around Lunada Bay, Muhs *et al*. (2006) estimated a ~60–70 m apsl elevation shoreline angle for the 'third' terrace of Woodring et al. (1946), an unnamed intermediate terrace between the Paseo del Mar (MIS 5a) and Gaffey (MIS 5e; the 'fifth' terrace of Woodring *et al*., 1946); as this terrace does

not have geochronological control, we do not include it as MIS 5c indicator.

### 5.1.13 San Joaquin Hills, California

Vedder *et al*. (1957) first mapped the emergent marine terraces at San Joaquin Hills. Grant *et al*. (1999) extended the mapping of previous unpublished surveys (see literature discussed therein) and revised the shoreline angle elevation of the first emergent terrace to 19 – 22 m apsl (Figure 2). Based on a single uranium series date on coral, and correlations between

terrace height and presumed eustatic sea level, Grant *et al*. (1999) argued that either the first terrace formed during MIS 5a and hosts reworked MIS 5c coral, or that the MIS 5a and 5c high stands occupied the same terrace (Figure 4).



### 5.1.14 San Nicolas Island, California

Vedder and Norris (1963) mapped 14 emergent bedrock-incised marine terraces on San Nicolas Island and documented their associated faunal assemblages. Muhs *et al.* (1994; 2006) reported 56 uranium-series dates on corals found in multiple

outcroppings of the lowermost two terraces (terraces 1 and 2) which assign these to MIS 5a and 5e, respectively (Figure 4). Muhs *et al.* (2012) utilized differential GPS measurements to revise previously reported shoreline angle elevations for terraces 1 and 2 and subdivided terrace 2 into two distinct geomorphic units (terraces 2a and 2b). Muhs *et al.* (2012) argued that, together, the geomorphic relationships and the 65 new uranium-series dates on solitary corals supported the conclusions that: (i) terrace 2a (36–38 m apsl) formed during MIS 5e; (ii) terrace 2b (28–33 m apsl), which hosts corals of ~120 and

~100 ka age clusters, formed during MIS 5c and captures reworked fossils from the adjacent, formerly more extensive 5e terrace (terrace 2a); and (iii) terrace 1 (8–13 m apsl) formed during MIS 5a. Muhs *et al.* (2012) posited that the distinctive faunal assemblages of terraces 1, 2a and 2b serve to further support their different ages. The terrace elevations reported by Muhs *et al.* (2012) are shown in Figures 2 and 3; Figures 4 and 5 show the full suite of uranium-series dates reported for San Nicolas Island terraces 1, 2a, and 2b (Muhs *et al.*, 1994; 2006; 2012).

### 5.1.15 Point Loma and Oceanside, California

Ellis (1919) first mapped five marine terraces at Point Loma. Hertlein and Grant (1944) briefly revisited the lower terraces. Carter (1957) extended the mapping to include an additional terrace cropping out lower than the lowest terrace surveyed by Ellis (1919), later named Bird Rock terrace (Kern, 1977). Kern (1973) documented deformation of the Point Loma terraces. Kern and Rockwell (1992) utilized hand levels to revise the shoreline angle elevation of the Bird Rock terrace to 9 – 11 m

apsl (Figure 2). The outcropping of Bird Rock terrace to the west at Oceanside is also mapped at 9 – 11 m apsl (Figure 2). Ku and Kern (1974) reported two uranium-series dates on mollusks but did not utilize these to assign an age to Bird Rock terrace due to secondary uptake of uranium in the mollusk shells. In the absence of radiometric dates, calibrated amino acid ratios support a Bird Rock terrace age assignment to MIS 5a (Kern 1977; Kern and Rockwell, 1992; Figure 4).

### 5.1.16 Malibu, California

Davis (1932) mapped three emergent marine terraces around Malibu, California, which, in ascending order, are named the Monic, Dume, and Malibu terraces, for which we focus on the westward tilting Dume terrace. Birkeland (1972) revisited the Dume terrace, of which the shoreline angle sits between ~7 and 40 m apsl apsl (Figure 3). Szabo and Rosholt (1969) reported seven uranium series dates on mollusks sampled from the Dume terrace consistent with an MIS 5c age (Figure 5), though these dates utilized an open-system model to compensate for mobile uranium within shells. If correct, these radiometric ages

imply that the Dume terrace correlates with San Nicolas Island terrace 2 (Birkeland 1972; see above). Szabo and Rosholt (1969) and Birkeland (1972) documented a higher, adjacent terrace referred to as terrace C/Corral terrace, which crops out

between the Malibu and Dume terraces. Szabo and Rosholt (1969) utilized the same corrected uranium-series methods to assign the Corral terrace to MIS 5e.

### 5.1.17 Punta Banda, Mexico

Lindgren (1889) first documented Punta Banda marine terraces and Allen *et al.* (1960) and Rockwell *et al.* (1989) mapped the lowest 13 and 12 terraces in detail, respectively. The Lighthouse terrace, the lowest mapped terrace (15 – 17.5 m apsl shoreline angle; Figure 2), is well preserved on the south side of the peninsula, though discontinuous on the north side (Rockwell *et al.*, 1989). Fifteen uranium-series dates on *Balanophyllia elegans* assign the Lighthouse terrace to MIS 5a, and extralimital northern species in the faunal assemblage support this designation (Rockwell *et al.,* 1989; Figure 4). A

fragmented, narrow second terrace crops out across the Punta Banda peninsula with a shoreline angle of 22 m apsl (Rockwell *et al.,* 1989; Figure 3). No radiometric dates exist for the second terrace, though Muhs et al. (1988) used stratigraphic relationships to infer an age of MIS 5c (Figure 5).

### 5.1.18 Punta Cabras, Mexico

At Punta Cabras, Baja California, Mexico, Addicott and Emerson (1959) first documented a narrow, discontinuous marine

terrace with inner edge elevations of 4.5 – 17 m apsl (Figure 2). No radiometric dates exist for this terrace, though a limited number of radiocarbon dates and extralimital northern species in the faunal assemblage of overlying marine and non-marine deposits support a terrace assignment to MIS 5a (Addicott and Emerson, 1959; Mueller et al., 2009; Figure 4).

### 5.1.19 Summary

The sea level indicators found on the west coast of North American consist primarily of emergent flights of wave-cut marine

terraces. Terraces assigned to MIS 5e and 5a are both present at many field localities; a minority of sites include a purported MIS 5c terrace bounded above and below, respectively, by an 5e and 5a terrace. Late Quaternary tectonic deformation affects the majority of west coast terrace exposures (e.g., McInelly and Kelsey), thus all present day terrace elevations must be corrected for tectonics before use in paleo-sea level reconstructions (e.g., Creveling et al., 2015; Simms et al., 2015). Differing vertical displacement across a single field locality complicates this correction and can result in a broad range and

terrace elevations with large uncertainties. Overall, MIS 5 substage terraces across this region overall have very good chronologies, particularly those hosting solitary corals suitable for uranium-thorium dating. These radiometric chronologies are supported by the widespread use of calibrated amino acid ratios and stratigraphic relationships.



### 5.2 East Coast of North America & the Caribbean

### 5.2.1 Virginia Beach, Virginia; Moyock, North Carolina; Charleston, South Carolina; and Skidaway, Georgia

Cronin *et al.* (1981) mapped emergent coral terraces along the east coast of the United States, from Virginia to Georgia, and reported reconstructions of paleo-sea level based upon their documented terrace elevations; Wehmiller *et al.* (2004) revisited the Virginia Beach, Moyock, Charleston and Skidaway sites, and reported the maximum elevation of the four coral bearing units as ~7.5 m apsl, ~5 m apsl, ~5 m apsl and ~3 m apsl respectively (Figure 2).  27 uranium-series dates on coral assign the terraces at these four sites to MIS 5a (Cronin *et al.,* 1981; Szabo, 1985; Wehmiller *et al.,* 2004; see Figure 4). Further, five

electron spin resonance dates and eight amino acid ratios at the Virginia Beach site support an age assignment of MIS 5 (Mirecki *et al.*, 1995). No MIS 5c-equivalent coral terraces were recognized across this region.

### 5.2.2 Pamlico Sound, North Carolina

Parham *et al.* (2013) utilized outcrops and sediment cores to map late Quaternary beach deposits, for which we focus on a deposit primarily composed of a thin veneer of sand and laminated sand. The sequence appears discontinuously through the

study site at an elevation of 9 – 14 m apsl (Figure 2): east of the Chowan river, the deposit forms a prograding spit whereas further east of the Suffolk shoreline the deposit forms a seaward thickening wedge, and the deposit is not well preserved in the northern and southern portions of the study area. Shelly marine material within the deposit was assigned to MIS 5a using calibrated amino acid racemization and optically stimulated luminescence dating (Figure 4).

### 5.2.3 Freeport Rocks, Texas

Simms *et al.* (2009) documented a sedimentary deposit within offshore sediment cores, referred to as the Freeport Rocks Bathymetric High, consisting of barrier island facies. The deposit appears at its highest elevation within the core at 18.9 m bpsl (below present day sea level; Figure 2). The authors assign the Freeport Rocks Bathymetric High to MIS 5a based upon a single optically stimulated luminescence date (Figure 4).

### 5.2.4 Fort St. Catherine, Bermuda

A rich literature describes the evolution of the stratigraphic nomenclature of Bermuda, comprised of six carbonate units separated by *terra rosa* paleosols (Land *et al.*, 1967; Vacher and Hearty, 1989; Hearty, 2002). The marine member of the Southampton Formation, mapped at 1 m apsl (Vacher and Hearty, 1989), was tentatively assigned to MIS 5a or 5c using amino acid stratigraphy (Harmon *et al.*, 1983; Hearty *et al.,* 1992) though, subsequently, 24 uranium-series dates reported across multiple studies have indicated an MIS 5a age assignment (Harmon *et al.*, 1983; Ludwig *et al.,* 1996; Muhs *et al.*,

2002; see Figure 4). The Pembroke unit of the Rocky Bay Formation, previously classified as the Pembroke Formation (with a proposed alternate name of the Hungry Bay Formation), has generated more debate. Amino acid ratios on *Poecilozonites* support a MIS 5c age assignment (Harmon *et al.*, 1983), while whole rock amino acid ratios support an age assignment of

MIS 5e (Hearty *et al.*, 1992). Four uranium-series dates yielded one MIS 5e-aged coral, two MIS 5c-aged corals, and one modern coral (Harmon *et al.*, 1983; see Figure 5) though Vacher and Hearty (1989) argued that published uranium-series

dates were unable to resolve an MIS 5 substage for the Rocky Bay Formation, and instead hypothesized that this unit represent MIS 5e. For the purposes of this review, the Pembroke unit chronology is included in the MIS 5c specific Fig. 5, with both 5c and 5e age assignments included. No elevation was reported for the Pembroke unit.

Moreover, controversy exists over the classification of both the Southampton Formation and the Pembroke unit of the Rocky Bay Formation. Harmon *et al.* (1983) hypothesized that both units formed from storm wave activity and, thus, the deposits

do not serve as indicators of a sea level high stand. Toscano and Lundberg (1999) supported this hypothesis, further specifying that the units were formed in the Holocene and incorporated corals of many different ages. Alternatively, other studies argued the location of Fort St. Catherine relative to the platform margin and the narrow range of MIS 5a dates on corals found within the Southampton Formation reveal that these units represent a sea level high stand (Vacher and Hearty, 1989; Ludwig *et al.*, 1996; Muhs *et al.*, 2002). Here we include both units within the database.

### 5.2.5 Sand Key Reef, Florida

Seismic-reflection profiles documented submerged outlier reefs along the windward side of the modern Florida Keys (Lidz *et al.*, 1991; Ludwig *et al.*, 1996; Toscano and Lundberg, 1999). While the Sand Key Reef shows geomorphic complexity, the primary reef crest sits ~10 – 12 m bpsl (Figure 2). The reef primarily comprises *Montastrea annularis*, with a thin overgrowth of reef crest *Acropora palmata* (Ludwig *et al.*, 1996). Two radiocarbon ages along with eight uranium-series

dates on *Montastrea annularis*, *Acropora palmata*, and *Colpophyllia natans* assign the main reef growth to MIS 5a (Lidz *et al.*, 1991; Ludwig *et al.*, 1996; Toscano and Lundberg, 1999; Figure 4).

### 5.2.6 Eleuthera Island, Bahamas

Skeletal eolianite comprising Eleuthera Island are interpreted as eolian dunes and are separated from the underlying formations by paleosols (Kindler and Hearty, 1996; Hearty, 1998; Hearty and Kaufman, 2000). The authors report an

inferred paleo-sea level of 0 – 5 m bpsl, rather the modern elevation of the deposit (Hearty and Kaufman, 2000; Figure 2). The stratigraphic position, location, and amount of diagenetic alteration of the eolianite, along with whole rock amino acid ratios, assign the unit to MIS 5a (Figure 4).

### 5.2.7 Berry Islands, Bahamas

Newell (1965) reported a single uranium-series date for a coral welded with caliche to a platform on Berry Island, which

assigned the coral to MIS 5a. Neumann and Moore (1975) reported 11 additional uranium-series dates on corals found at 0.2 – 2.3 m apsl (Figure 3) as MIS 5 in age, though the range of ages could not assign the ridge to a specific substage high stand. A later review noted that an MIS 5c age assignment for the Berry Islands samples fit the uncertainty of the published age data (Creveling *et al.,* 2017; Figure 5).





### 5.2.8 Northwestern Peninsula, Haiti

Woodring *et al.* (1924) first documented the geology of the Northwestern Peninsula, which was followed by further research summarized by Dodge *et al.* (1983). Dodge *et al.* (1983) mapped seven constructional coral reef terraces composed primarily of *Acropora palmata*, for which we focus on the lowest two, the Mole and Saint terraces. Dumas *et al.* (2006) published altimeter surveys for the Mole and Saint terraces (also referred to as T1 and T2), revising the previously reported inner edge elevations to 23 m apsl and 37 m apsl, respectively (Figures 2 and 3). Dodge *et al.* (1983) and Dumas et al. (2006) argued

that 15 uranium-series dates on corals support the conclusion that: (i) the Mole terrace formed during MIS 5a; (ii) the Saint terrace formed during MIS 5c; (iii) both terraces host corals reworked from the adjacent MIS 5e terrace (see Figures 4 and 5).

### 5.2.9 Christ Church and Clermont Nose traverses, Barbados

Mesolella (1967) first mapped the coral reef terraces of the Christ Church and Clermont Nose traverses. Bender et al. (1979)

revisited nine reef tracts at Christ Church and seven reef tracts at Clermont Nose. 23 uranium-series dates on coral, along with 12 Protactinium-231 dates, assign the Worthing and Ventor terraces at both traverses to MIS 5a and 5c respectively (Broecker *et al.*, 1968; Mesolella, 1969; Bender *et al.*, 1979; Edwards *et al.*, 1997; see Figures 4 and 5). The elevations of the Worthing and Ventor terraces are mapped at 3 m apsl and 6 m apsl at Christ Church and 20 m apsl and 30 m apsl at Clermont nose (Figures 2 and 3).

### 5.2.10 Summary


The east coast of North America and Caribbean sites consist of constructional coral reef terraces and terrestrial eolianites. Most field sites host only an MIS 5a sea level indicator and the common absence of adjacent MIS 5e indicators prevents the assessment and application of a tectonic uplift correction (see discussion in Potter and Lambeck, 2003). Given the abundance of amenable carbonate samples within the predominately reef and marine terrace indicators, high precision chronologies

have been developed based on uranium-thorium dating. The chronologies for eolianite indicators tend to be less precise and reproducible, in part due to allochthonous (reworked) nature of the dated corals from adjacent lithostratigraphic units.

### 5.3 Far Field

### 5.3.1 Oahu, Hawaii

Sherman et al. (2014) utilized offshore drill cores to document and date two submerged coral reef terraces on Oahu. A single

uranium-thorium date assigned one terrace to MIS 5a; two uranium-thorium dates assigned a second terrace to MIS 5c (Figures 4 and 5), though distinct elevations were not reported for the MIS 5a and 5c terraces. Instead, a range in elevations from 20 – 30 m bpsl was provided for both terraces (Figures 2 and 3).

### 5.3.2 Bahía Inglesa, Chile

Marquardt et al. (2004) utilized altimeter surveys to map eight emergent marine terraces and the overlying fossiliferous
terrace deposits. The shoreline angle of the two terrace lowest terraces found above the Holocene beach are mapped at 10 m
apsl and 31 m apsl (Figures 2 and 3). Uranium-series and electron spin resonance dates assign the two terraces to MIS 5, and
their stratigraphic position, assuming uniform uplift rate, further assigns the 10 m and 31 m terraces to MIS 5a and 5c (see
discussion in Marquardt et al., 2004; Figures 4 and 5).

### 5.3.3 Daebo-Gori region, Korea

Choi *et al.* (2008) utilized differential GPS to map wave-cut terraces in the Daebo-Gori region. The T2 terrace, with a
shoreline angel mapped at 8 – 11 m apsl (Figure 2), is a wide and continuous surface, sparsely covered in sediment of up to 2
m thick; the T3b terrace, with a shoreline angle mapped at 17 – 22 m apsl (Figure 3), is a narrow, lower bench of the T3
terrace, separated from the upper bench by a gently sloping riser, which in places makes it difficult to distinguish the two
platforms. 16 optically stimulated luminescence (OSL) dates assign the T2 terrace to MIS 5a (Choi *et al.*, 2003; see literature
referenced in table 2 of Choi et al., 2008; see Figure 4); eight OSL dates and two paleomagnetic dates assign the T3b terrace
to MIS 5c (Choi et al., 2003; Choi et al., 2008; see literature referenced in table 3 of Choi et al., 2008; see Figure 5).

### 5.3.4 Pamilacan Island, Philippines

Ringor *et al.* (2004) mapped the coral reef terraces on Pamilacan Island, for which four uranium-series dates on corals have
assigned the unnamed broad terrace, with a shoreline angle mapped at 6 m apsl, and the more narrow and poorly developed
terrace (also unnamed), with a shoreline angle mapped at 13 m apsl, to MIS 5a and 5c.

### 5.3.5 Panglao Island, Philippines

Omura *et al.* (2004) mapped the coral reef terraces found along Panglao Island. The lowest well-developed terrace is mapped
at 5 m apsl (Figure 3) and is composed of two geomorphic units. The lower unit contains one uranium-series date on Porites
which dates to late MIS 5e. Four uranium-series dates on Platygyra ryukyuensis assign the upper terrace, of which the inner
margin is mapped at 5 m apsl, to MIS 5c (Figure 5).

### 5.3.6 Tewai and Kwambu Sections, Huon Peninsula

Chappell (1974) first mapped the emergent coral reef terraces along the northern coast of the Huon Peninsula, for which we
focus on the well preserved, laterally tilted Va and VIa terraces. 11 uranium-series dates on corals and mollusks assign these
terraces to MIS 5a and 5c (Veeh and Chappell, 1970; Chappell, 1974; Esat et al., 1999; Cutler et al., 2003; Figures 4 and 5).
Chappell and Shackleton (1986) reported the reef crest elevations of the Va and VIa reef terraces to be 260 m apsl and 338 m
apsl at the Tewai section and 117 m apsl and 160 m apsl at the Kwambu section (Figures 2 and 3).

### 5.3.7 Hateruma Island, Japan

Ota and Hori (1980) proposed an initial chronology for the six lowest Quaternary terraces around Hateruma Island following early mapping efforts discussed in Ota and Omura (1992). Omura (1984) published four uranium-series dates on coral that

assigned terraces V and IV to MIS 5a and 5c, respectively (Figures 4 and 5). The updated elevation surveys of Ota and Omura (1992) mapped the maximum height of the terraces V and IV at 23 m apsl and 30 m apsl, respectively (Figures 2 and 3).

### 5.3.8 Atauro Island, East Timor

The lowest two coral reef terraces of Atauro Island, termed 1a and 1b, are mapped at inner edge elevations of 20 m apsl and

36 m apsl, respectively (see literature discussed in Chappell and Veeh, 1978; Chappell and Veeh, 1978; Figures 2 and 3). Two uranium-series dates on corals have assigned terrace 1b to MIS 5c (Figure 5). While no radiometric dates exist for terrace 1a, Chappell and Veeh (1978) used the position of this terrace on age-height plots to infer an age of MIS 5a (Figure 4).

### 5.3.9 Spencer Gulf, Australia

Hails *et al.* (1984) utilized sediment cores and sonar to study shallow water stratigraphy in the Spencer Gulf, for which we focus on the Lowly Point and False Bay formations. The Lowly Point and False Bay formations occur at water depths exceeding 14 m bpsl and 8 m bpsl, respectively (Figures 2 and 3). No radiometric dates exist for either formation; the stratigraphic position of the Lowly Point and False Bay formations between the MIS 5e age Mambray formation and the Holocene age Germein formation (see literature discussed in Hails et al., 1984), along with the extent to which the

depositional events flooded Spencer Gulf, support age assignments of MIS 5a and 5c (Figures 4 and 5).

### 5.3.10 Robe Range, Australia

Sprigg (1952) first mapped the eolianite barrier-shoreline complex of the Robe Range for which we focus on the Robe III unit. Murray-Wallace (2002) reported the shoreline elevation of Robe III as 2 m bpsl (Figure 3). A single luminescence date, in conjunction with the stratigraphic position of the Robe III unit, supports an age assignment of MIS 5c (Huntley et al.,

1994; Figure 5).

### 5.3.11 Makran Subduction Zone, Iran (Lipar, Ramin, Gurdim, Jask sites)

Harrison (1941) first documented the marine terraces along the Makran subduction zone of Iran; subsequent authors extended the initial mapping, documenting the differential uplift and lateral tilting of the terraces (Page et al., 1979; Reyss et al., 1998; Normand et al., 2019). Normand et al. (2019) revisited the terraces and provided updated elevations for the

terraces found at the Lipar, Ramin, Gurdim, Jask sites. The majority of the T1 terraces, dated with three optically stimulated



luminescence dates and a single uranium-series date to MIS 5a (Figure 4), have a shoreline angle mapped at 2 m bpsl to 65 m apsl (Normand et al., 2019; Figure 2). Normand et al. (2019) expanded on the history of the T1 terrace at the Ramin site, which hosts two luminescence samples dated to MIS 5e and 5a, respectively, concluding that the chronology implies a reoccupation of the T1 terrace during multiple sea level high stands (Figure 4). The Lipar site also contains a T2 terrace

mapped at 45 m apsl (Figure 2) which hosts a luminescence sample dated to MIS 5a (Figure 4).

### 5.3.12 The Gulf of Gabes, Tunisia

Gzam et al. (2016) utilized high precision echo sounders to map two submerged beach ridges, with peak elevations of 8 m and 19 m bpsl (Figure 2 and 3). From their analysis of biocalcarenite development, the authors concluded that the ridge formation could indicate a rapid sea level transgression which, along with the stratigraphic position of the 8 m and 19 m

ridges, supported their respective age assignments of MIS 5a and 5c (Figure 4 and 5). At present, no radiometric dates exist for these beach ridges.

### 5.3.13 Summary

Relative sea-level indicators cropping out in the far field include uplifted marine and coral reef terraces and submerged sedimentary packages. While the chronologies for some sites are based on uranium-thorium dating of carbonate samples, age

assignments based on luminescence dating are more commonly used at far field sites than in the other regions. Two submerged sites rely solely on stratigraphic relationships to assign an MIS substage to the respective sea level indicators. Given the range of methods applied, the quality of chronologies varies by site and indicator.



**Table 1: ᵃSLI = Sea Level Indicator, TL = Terrestrial Limiting, ML = Marine Limiting, ᵇTerrace chronology does not differentiate MIS 5 substage. Summary of MIS 5a RSL Indicator data contained in review.**



| Marine Isotope Stage 5a | | | | | | | |
|---|---|---|---|---|---|---|---|
| **Site** | **Latitude** | **Longitude** | **Elevation** | **Indicator Type[a]** | **Dating Method** | **Indicator Quality** | **Age Quality** |
| Newport | 44.63 | -124.05 | (0 - 45) ± 3.3 | SLI | AAR, Other | 1 | 2 |
| Cape Arago | 43.306531 | -124.401657 | 31 ± 2 | SLI | Other | 3 | 2 |
| Coquille Point | 43.114117 | -124.437096 | 18 ± 2 | SLI | U-Series, AAR | 3 | 4 |
| Brookings | 42.05 | -124.28 | (30 - 62) ± 6 | SLI | Other | 1 | 2 |
| Bruhel Point | 39.607469 | -123.786856 | 10 ± 2 | SLI | Stratigraphy, AAR | 3 | 2 |
| Point Reyes | 37.9 | -122.6938 | (38 - 77) ± 3.4 | SLI | Luminescence | 1 | 4 |
| Santa Cruz (Western terrace) | 36.96 | -122.09 | 87 ± 3.3 | SLI | Other | 2 | 3 |
| Santa Cruz (Davenport terrace) | 37.0294755 | -122.1923133 | (5 - 10) | SLI | U-Series, AAR, Other | 1 | 4 |
| San Simeon[b] | 35.639462 | -121.1848936 | (5 - 24) ± 2 | SLI | U-Series, Luminescence, Other | 1 | 3 |
| Point Buchon | 35.2552529 | -120.8990702 | (5 - 14) ± 2 | SLI | U-Series, Other | 1 | 1 |
| Gaviota | 34.4675398 | -120.2674989 | (10 - 17) ± 2 | SLI | U-Series, AAR, Other | 3 | 1 |
| Santa Rosa Island | 34.003812 | -120.195602 | (5.4 - 7.4) ± 1 | SLI | AAR | 3 | 2 |
| San Miguel Island | 34.0191768 | -120.3177209 | 3.5 | SLI | Stratigraphy | 1 | 2 |
| Palos Verdes Hills | 33.7246996 | -118.3552086 | (45 - 47) | SLI | U-Series, AAR | 1 | 4 |
| San Joaquin Hills | 33.5696784 | -117.8385408 | (19 - 22) | SLI | U-Series | 1 | 3 |
| San Nicolas Island | 33.2472453 | -119.5070695 | (8 - 13) ± 0.3 | SLI | U-Series | 3 | 4 |
| Oceanside | 33.17 | -117.35 | (9 - 11) ± 2 | SLI | AAR | 3 | 2 |
| Point Loma | 32.67 | -117.24 | (9 - 11) ± 2 | SLI | U-Series, AAR | 3 | 3 |
| Punta Banda | 31.7455541 | -116.7394247 | (15 - 17.5) ± 0.2 | SLI | U-Series | 3 | 4 |
| Punta Cabras | 31.33 | -116.44 | (4.5 - 17) | SLI | Other | 1 | 3 |
| Virgina Beach | 36.7823 | -76.1966 | 7.5 ± 3 | SLI | U-Series, AAR, ESR | 3 | 3 |
| Moyock | 36.508 | -76.153 | 5 ± 3 | SLI | U-Series | 3 | 4 |
| Pamlico Sound | 35.4828287 | -75.951469 | (9 - 14) | SLI | Luminescence | 1 | 4 |
| Charleston | 32.8586 | -79.7803 | 5 ± 3 | SLI | U-Series | 3 | 4 |
| Skidaway | 31.916 | -81.071 | 3 ± 3 | SLI | U-Series | 3 | 4 |
| Freeport Rocks | 27.7852552 | -96.9132876 | -18.9 | TL | Luminescence | 1 | 2 |
| Fort St. Catherine | 32.390733 | -64.674732 | (1 - 2) ± 1 | TL | U-Series, AAR | 0 | 4 |





| Sand Key Reef | 24.4977811 | -81.8477838 | (-12 - -10) | SLI | U-Series | 1 | 4 |
|---|---|---|---|---|---|---|---|
| Eleuthera Is. | 25.0090818 | -76.3780482 | (-5 - 0) | SLI | AAR | 1 | 2 |
| Northwestern Peninsula | 19.8039797 | -73.3097345 | 23.4 | SLI | U-Series | 1 | 4 |
| Christ Church | 13.0699141 | -59.5693843 | 3 +1/-2 | SLI | U-Series, Other | 3 | 4 |
| Clermont Nose | 13.1350465 | -59.6345479 | 20 +3/-2 | SLI | U-Series, Other | 3 | 3 |
| Oahu | 21.4477881 | -158.1997812 | (-30 - -20) | SLI | U-Series | 1 | 2 |
| Bahía Inglesa | -27.1142508 | -70.8742622 | 10 ± 5 | SLI | Stratigraphy | 2 | 2 |
| Daebo-Gori Region | 36.0554812 | 129.5444388 | (8 - 11) | SLI | Luminescence | 1 | 2 |
| Pamilacan Island | 9.4951118 | 123.9281264 | 6 ± 1 | SLI | U-Series | 3 | 4 |
| Tewai Section | -6.2206748 | 147.6766172 | 260 | SLI | U-Series, Stratigraphy | 1 | 4 |
| Kwambu Section | -6.0704502 | 147.5171788 | 117 | SLI | U-Series, Stratigraphy | 1 | 3 |
| Hateruma Island | 24.058648 | 123.7817205 | 23 ± 2 | SLI | U-Series | 3 | 3 |
| Atauro Island | -8.3012532 | 125.5565624 | 20 | SLI | Other | 1 | 2 |
| Lipar | 25.2567922 | 60.80393 | 20 ± 2.1 | SLI | U-Series, Luminescence | 3 | 2 |
| Lipar | 25.2591769 | 60.7979098 | 45 ± 2.1 | SLI | Luminescence | 3 | 2 |
| Ramin | 25.2694195 | 60.763467 | -2 ± 4.3 | SLI | Luminescence | 2 | 2 |
| Gurdim | 25.3390736 | 60.1665682 | 62 ± 2.1 | SLI | Luminescence | 3 | 2 |
| Jask | 25.6552358 | 57.7874395 | -3 ± 4.3 | SLI | Luminescence | 2 | 2 |
| Bsissi | 33.7171283 | 10.313814 | -8 ± 1 | TL | Stratigraphy | 0 | 2 |
| Ghannouche | 33.7072534 | 10.3343664 | -9 ± 1 | TL | Stratigraphy | 0 | 2 |
| Teboulbou | 33.7007635 | 10.3519874 | -8 ± 1 | TL | Stratigraphy | 0 | 2 |
| Kettana | 33.6785967 | 10.4125647 | -9 ± 1 | TL | Stratigraphy | 0 | 2 |
| Zarat | 33.6785967 | 10.4125647 | -8 ± 1 | TL | Stratigraphy | 0 | 2 |
| Zerkine | 33.6785967 | 10.4125647 | -18 ± 1 | TL | Stratigraphy | 0 | 2 |
| Spencer Gulf | -33.9120079 | 136.8615574 | -8 | ML | Other | 0 | 2 |


**Table 2: ᵃSLI = Sea Level Indicator, TL = Terrestrial Limiting, ML = Marine Limiting**

| Marine Isotope Stage 5c | | | | | | | |
|---|---|---|---|---|---|---|---|
| **Site** | **Latitude** | **Longitude** | **Elevation** | **Indicator** | **Dating Method** | **Indicator** | **Age Quality** |





| | | | | Type[a] | | Quality | |
|---|---|---|---|---|---|---|---|
| Newport | 44.63 | -124.05 | (0 - 85) ± 6 | SLI | Other | 1 | 2 |
| Cape Arago | 43.306531 | -124.401657 | 68 ± 2 | SLI | Stratigraphy | 3 | 2 |
| Coquille Point | 43.114117 | -124.437096 | 70 ± 2 | SLI | Stratigraphy | 3 | 2 |
| Brookings | 42.05 | -124.28 | (57 - 90_ ± 6 | SLI | Other | 1 | 2 |
| Bruhel Point | 39.607469 | -123.786856 | 23 ± 2 | SLI | Stratigraphy | 3 | 2 |
| Santa Cruz (Wilder terrace) | 36.96 | -122.09 | 132 ± 3.3 | SLI | Other | 2 | 3 |
| Santa Cruz (Highway 1 terrace) | 37.0294755 | -122.1923133 | 26 | SLI | Other | 1 | 2 |
| Gaviota | 34.4675398 | -120.2674989 | 25 ± 5 | SLI | AAR, Other | 2 | 2 |
| San Nicolas Island | 33.2472453 | -119.5070695 | (28 - 33) ± 0.3 | SLI | U-Series | 3 | 3 |
| Malibu | 34.03 | -118.71 | (7.6 - 39.6) | SLI | U-Series | 1 | 4 |
| Punta Banda | 31.7455541 | -116.7394247 | 22 ± 0.2 | SLI | Stratigraphy | 3 | 2 |
| Fort St. Catherine | 32.390733 | -64.674732 | Not Reported | TL | U-Series, AAR | 0 | 3 |
| Berry Island | 25.6250042 | -77.8252203 | (0.2 - 2.3) ± 1 | SLI | U-Series | 3 | 3 |
| Northwestern Peninsula | 19.8039797 | -73.3097345 | 37.2 | SLI | U-Series | 1 | 3 |
| Christ Church | 13.0699141 | -59.5693843 | 6 +5/-1 | SLI | U-Series, Other | 3 | 4 |
| Clermont Nose | 13.1350465 | -59.6345479 | 30 +3/-2 | SLI | U-Series, Other | 3 | 4 |
| Oahu | 21.4477881 | -158.1997812 | (-30 - -20) | SLI | U-Series | 1 | 4 |
| Bahía Inglesa | -27.1142508 | -70.8742622 | 31 ± 5 | SLI | Stratigraphy | 2 | 2 |
| Daebo-Gori Region | 36.0554812 | 129.5444388 | (17 - 22) | SLI | Luminescence, Other | 1 | 3 |
| Pamilacan Island | 9.4951118 | 123.9281264 | 13 ± 1.6 | SLI | U-Series | 3 | 4 |
| Panglao Island | 9.573798 | 123.8221394 | 5 | SLI | U-Series | 1 | 4 |
| Tewai Section | -6.2206748 | 147.6766172 | 338 | SLI | Stratigraphy | 1 | 2 |
| Kwambu Section | -6.0704502 | 147.5171788 | 160 | SLI | U-Series, Stratigraphy | 1 | 3 |
| Hateruma Island | 24.058648 | 123.7817205 | 30 ± 2 | SLI | U-Series | 3 | 2 |
| Atauro Island | -8.3012532 | 125.5565624 | 36 | SLI | U-Series | 1 | 4 |
| Zarat | 33.6785967 | 10.4125647 | -18 ± 1 | TL | Stratigraphy | 0 | 2 |
| Zerkine | 33.6785967 | 10.4125647 | -20 ± 1 | TL | Stratigraphy | 0 | 2 |
| Spencer Gulf | -33.9120079 | 136.8615574 | -14 | ML | Other | 0 | 2 |
| Robe Range | -37.219789 | 139.787838 | -2 | TL | Luminescence | 1 | 2 |



## 6 Future Research Directions

The global database of sea level indicators dated (or assigned) to MIS 5a and 5c presented in this paper densely covers the Pacific coast of North America (18 field sites), the Atlantic coast of North America and the Caribbean (9 field sites), and more sparsely covers the remaining globe (12 field sites). The broad geographic spread of the data allows for an increasingly resolved reconstruction of MIS 5a and 5c GMSL sea level, with especially good coverage in the near field of the North American Ice Sheets. Future research directions could address scant indicators reported (in English language journals)
outside of North America, and refine radiometric chronologies for existing indicators.

## 7 Data Availability

The database detailed in this study is available at at https://doi.org/10.5281/zenodo.4426206 (Thompson and Creveling, 2021). The content at this link were exported from the WALIS database interface on 7 January 2020. A summary of these WALIS data can be found in Tables 1 and 2 of this manuscript. Description of each data field in the database is contained at
this link: https://doi.org/10.5281/zenodo.3961543 (Rovere et al., 2020). More information on the World Atlas of Last Interglacial Shorelines can be found here: https://warmcoasts.eu/world-atlas.html. Users of this database are encouraged to cite the primary literature sources as well as this article.

**Author Contribution** S.T. assumed primary responsibility for all entries into the WALIS database, and the illustration of these data. S.T. extended the literature review of MIS 5a and 5c indicators beyond that reported in Creveling et al. (2017).
S.T. and J.R.C. contributed equally to the structure and writing of the manuscript.

**Competing Interests** The authors declare that they have no conflict of interest.

**Acknowledgments** The data presented in this publication were compiled in WALIS, a sea-level database interface, developed with funding from the ERC Starting Grant "WARMCOASTS" (ERC-StG-802414), in collaboration with
PALSEA (PAGES/INQUA) Working Group. The database structure was designed by A. Rovere, D. Ryan, T. Lorscheid, A. Dutton, P. Chutcharavan, D. Brill, N. Jankowski, D. Mueller, M. Bartz. E. Gowan and K. Cohen. The authors wish to thank Daniel Muhs for clarifying innumerable aspects of MIS 5a and 5c field observation and geochronological caveats over many years of conversation and Alessio Rovere for constructive comments on manuscript drafts and database entries.



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
