# Peer review of "A Global Database of Marine Isotope Substage 5a and 5c Marine Terraces and Paleoshoreline Indicators"

_Earth System Science Data, 2021_

## Author Response (AR1)

We thank Dr. Collin Murray-Wallace and the anonymous referee for the constructive reviews of our manuscript. Below we outline a comprehensive plan to respond to both referee's comments Below you will find the reviewer's text in bold, and our indented plain text responses with quotations (and appropriate line numbers) for proposed manuscript additions and edits.

--Schmitty Thompson and Jessica Creveling

**Referee [#]1 (Colin Murray-Wallace)**

**This is a very valuable contribution in presenting in one article, and in a global context, information about the spatial distribution and elevation of sedimentary successions correlating with the Late Pleistocene warm interstadials MIS 5c and 5a. In this synthesis, it is noted that one of the difficulties in reviewing the literature, is the veracity of some of the palaeosea-level indicators.**

**The manuscript would be enhanced by perhaps adopting a more critical analysis of some of the data and field relationships of the landforms and sediments described. The manuscript would also be improved by a stronger and perhaps more critical synthesis section at the end of the work, with more commentary on the role of GIA in explaining the inferred relative sea level observations, particularly for the US Atlantic Coastal Plain and some sectors of the Caribbean. While it is important to present the data and honour the data of our predecessors, it would be nice to see what the authors make of all the information (i.e. to put their 'stamp' on things in a synthesis/discussion section).**

> We thank the Referee for their supportive comments on our WALIS contribution. We chose to convey a critical analysis of the field data and relationships by elaborating on geological rationale for WALIS Quality Rating. We offer this critique for all three regional sub-sections:

> North American Pacific coast (revised manuscript lines 395 – 447):

"Marine terraces comprise the entirety of MIS 5a/5c relative sea level indicators cropping out along the North American Pacific coast. While in principle marine terraces can serve as excellent quality indicators (see 3 and 4), Pacific coast marine terraces receive quality ratings from very poor (1) to average (3) (Figure 6a,d; Tables 1 and 2). These ratings reflect four systematic uncertainties. First, no primary reference reports a terrace's indicative range, the distance between the storm wave swash height and the wave breaking depth (Vacchi et al., 2014; Rovere et al., 2016), and this precludes a calculation of indicative meaning. Given this absence, we used the IMCalc software to quantify the indicative meaning of all Pacific coast marine terraces (WALIS RSL IDs 3473–3503; Lorscheid and Rovere, 2019). The arising indicative ranges tend to vary between 4 and 7.5 m which necessitates quality ratings of poor (2) and lower (as for MIS 5a and 5c terraces at Cape Arago, Coquille Point, Bruhel Point, Gaviota, and San Nicolas Island; MIS 5a terraces at Santa Cruz (Western and Davenport), San Miguel Island, Santa Rosa Island, Oceanside, and Point Loma; and the MIS 5c terrace at Santa Cruz (Wilder)). The MIS 5a and 5c terraces at Punta Banda are the sole exception, as these elevation measurements are sufficiently precise to

warrant average quality (3) ratings. Second, the primary literature rarely reports the reference datum for the RSL indicator elevation. Third, the use of altimetry and topographic map measurement techniques before the widespread adoption of differential GPS means that literature-reported measurement uncertainties generally exceed ~3 m (as for the MIS 5a terraces at Palos Verdes Hills and San Joaquin Hills). Fourth, many regional terraces also crop out over a range of elevations due to faulting or tilting, and this range further contributes to RSL uncertainty (such as for MIS 5a and 5c terraces at Newport and Brookings; MIS 5a terraces at Point Reyes, San Simeon, Point Buchon, Punta Cabras; and MIS 5c terraces at Santa Cruz (Highway 1) and Malibu). As all four systematic uncertainties apply to many Pacific coast marine terraces, and at least one uncertainty applies to all terraces, the Pacific coast yields quality ratings of very poor (1) to average (3). From this we conclude that revisiting indicator elevation measurements with modern mapping methods could better constrain Pacific coast MIS 5a and 5c terrace elevations, indicative meanings, and RSL uncertainties.

We assigned the chronologies of the North American Pacific coast marine terraces ratings from good (4) to poor (2) (Fig. 7a,d; Tables 1 and 2). The two methods that conferred good quality (4) ratings for this region include: first, reproducible, high precision uranium-series ages on solitary coral skeletal carbonate (as for MIS 5a terraces at Coquille Point, Santa Cruz (Davenport terrace), Palos Verdes Hills, San Nicholas Island, and Punta Banda) and, second, high-precision luminescence ages (as for the MIS 5a Point Reyes marine terrace). For chronologic methods that yielded age uncertainty beyond the bounds of an MIS 5 substage—often arising from substrate experiencing open-system diagenesis—we applied an average (3) quality rating. Examples of this include uranium-series ages on coral (the MIS 5a terrace at San Joaquin Hills and the MIS 5c terrace at Malibu,) or molluscs (the MIS 5a terrace at Point Loma), cosmogenic ages (the MIS 5a and 5c terraces at Santa Cruz (Western and Wilder)), radiocarbon ages (the MIS 5a terrace at Punta Cabras), and luminescence ages (the MIS 5a terrace at San Simeon). While Muhs et al. (2012) interpreted the MIS 5c and 5e uranium-series ages on corals from San Nicholas Island Terrace IIb as an indication of terrace reoccupation, which could afford a good (4) quality rating, here we assign Terrace IIB an average (3) quality rating given that these ages span a time interval longer than either individual MIS substage. Poor quality (2) chronology ratings arise from relative dating methods, such as AAR on molluscs (the MIS 5a and 5c terraces at Gaviota, and the MIS 5a terraces at Newport, Bruhel Point, Santa Rosa Island, and Oceanside), and terrace counting (MIS 5a and 5c terraces at Bruhel Point; the MIS 5a terrace at San Miguel Island; and the MIS 5c terraces at Cape Arago, Coquille Point, and Punta Banda). For this region, chronologic assignment by terrace counting is especially common for MIS 5c terraces with an adjacent, well-dated MIS 5a terrace. Non-traditional methods and maximum/minimum limiting ages confer a poor (2) rating for MIS 5a and 5c terraces at Newport, Brookings, and Gaviota; the MIS 5a terrace at Cape Arago; and the MIS 5c terrace at Santa Cruz (Highway 1 terrace). Likewise, minimum limiting uranium-series age on mammal teeth and bone, and unreliable uranium-series dates on coral (see Hanson et al., 1992) assign Point Buchon a poor quality rating (2). For the North American Pacific coast, MIS 5a terraces generally received higher quality ratings than MIS 5c terraces (Tables 1 and 2)."

North American Atlantic coast and Caribbean (revised manuscript lines 548 – 580):

"The North American Atlantic coast and Caribbean sea level indicators, which consist of coral reef terraces, beach deposits, and terrestrial limiting beach ridges, received elevation quality ratings from good (4) to rejected (0) (Figure 6b,,e; Tables 1 and 2). Since none of the primary literature sources report indicative ranges for the relative sea level indicators along the North Atlantic coast and Caribbean, we calculated these values with the IMCalc software and reported these for WALIS RSL IDs 3504–3508, 3511–3519, 3556, and 3983–3984 (Lorscheid and Rovere, 2019). The clear reference datum for the Berry Island and MIS 5a Christ Church coral reef terraces (Neumann and Moore, 1975; Bender et al., 1979) warrants the only good quality (4) ratings for this region. For MIS 5a and 5c Clermont Nose and those coral reef terraces with no reference data reported (as for MIS 5a RSL indicators at Virginia Beach, Moyock, Charleston, Sand Key Reef, Eleuthera Island, and Southern Coast Barbados), the assigned quality ratings of average (3) reflect the precision of the elevation measurement reported in the primary literature and indicative ranges equal to, or less than, ~2 m as calculated by IMCalc. Coral reef terraces and beach deposits with a poor quality rating (2) reflect either a greater elevation measurement uncertainty (as for the MIS 5a and 5c RSL indicators at Northwestern Peninsula, the MIS 5a indicator at Pamlico Sound, and the MIS 5c indicators at Christ Church and Southern Coast Barbados) or an indicative range greater than ~2 as calculated by IMCalc (as for MIS 5a Skidaway). The terrestrial limiting MIS 5a indicator at Freeport Rocks and the MIS 5a and 5c indicators at Fort St. Catherine warrant a rejected (0) quality rating. As with the Pacific coast indicators, we support revisiting coral reef terraces and beach deposit with quality ratings of average (3) to poor (2) to improve uncertainty by applying modern methods for elevation measurement and adopting an updated framework to constrain indicative meaning.

We assigned the North American Atlantic Coast and Caribbean chronologies ratings of good (4) to poor (2) (Figures 7b,e; Tables 1 and 2). For this region, good ratings (4) arose from one of two methods: first, high precision uranium-series ages from skeletal coral (from MIS 5a and 5c indicators at Northwestern Peninsula, Christ Church, and Southern Coast; MIS 5a indicators at Virginia Beach, Moyock, Charleston, Skidaway, Sand Key Reef, and Fort St. Catherine; and the MIS 5c indicator at Clermont Nose) and second, high precision luminescence ages (on MIS 5a indicators at Pamlico Sound and Freeport Rocks). Sites that receive an average quality rating (3) have uranium-series based chronologies which span MIS 5 (the MIS 5a indicator at Clermont Nose; and the MIS 5c indicators at Fort St. Catherine and Berry Island). Due to the absence of material available for numerical dating methods at Eleuthera Island, the relative dating methods warrants a quality rating poor (2). We advocate continued focus on sites with average quality ratings (3)—those that have substrate amenable to geochronology yet have imprecise ages—to improve chronologic assignments."

Far field region (revised manuscript lines 684 – 718):

"Sea level indicators in the far field encompass marine terraces, coral reef terraces, and both marine and terrestrial limiting indicators (shallow water facies and beach ridges, respectively), which have quality ratings between average (3) and rejected (0) (Figures 6c,f; Tables 1 and 2). As with the previous two regions, we used IMCalc to quantify the indicative meaning of each relative sea level indicator in the absence of reported modern analog data (WALIS RSL IDs 3520–3537, 3557–3562, 3982; Lorscheid and Rovere, 2019). Most primary references do not report a clear reference datum, which limits the maximum quality rating to average (3). An average (3) rating for a coral reef terrace typically reflects both a precise measurement and a

narrow indicative range (less than ~2 m) as calculated by IMCalc (as for MIS 5a and 5c indicators at Pamilacan Island and Hateruma Island and the MIS 5c indicator at Panglao Island). In contrast, as marine terraces typically have larger indicative ranges, an average (3) quality rating necessitates a precise elevation measurement (e.g., the MIS 5a and 5c indicators at Daebo-Gori region and the MIS 5a indicators at Lipar and Gurdim). Poor ratings (2) for marine and coral reef terraces arise from imprecise measurements intrinsic to survey methods such as altimeters and topographic maps, or from not reporting measurement uncertainty (such as the MIS 5a and 5c indicators at Bahía Inglesa, Oahu, Atauro Island, Tewai Section, Kwambu Section and MIS 5a indicators at Ramin and Jask). All sites in Tunisia and Australia receive rejected (0) quality ratings because they only provide minimum/maximum limiting constraints on paleo-sea level. In summary, we advocate for applying modern measurement methods (with more precise error estimates) to marine and coral reef terraces with present quality ratings of average (3) and poor (2) to refine indicative meaning.

The far field chronologies quality ratings range from good (4) to poor (2) (Figures 7c,f; Tables 1 and 2). Good quality ratings (4) are conferred through two avenues: first, from uranium-series ages on skeletal coral reef terraces that fall within a single MIS substage (such as for MIS 5a and 5c indicators at Oahu and Pamilacan Island, the MIS 5a indicator at Tewai Section; and the MIS 5c indicators at Panglao Island, Hateruma Island and Atauro Island) and second, from replicated high precision luminescence ages (on MIS 5a indicators at Lipar, Gurdim, and Jask; and the MIS 5c indicator at Robe Range). Average ratings (3) reflect chronologies where the age uncertainty exceeds a single MIS substage, either through uranium-series dating on coral (as for the MIS 5a and 5c Kwambu Section indicators and the MIS 5a Hateruma Island indicator) or luminescence dating (the MIS 5a Ramin indicator). A poor rating (2) may reflect one of a number of relative dating methods used, including: minimum limiting luminescence and paleomagnetic ages (MIS 5a and 5c Daebo-Gori indicators), terrace counting (indicators at MIS 5a and 5c Bahía Inglesa, Zarat, Zerkine, and Spencer Gulf; MIS 5a indicators at Bissi, Ghannouche, Teboulbou, and Kettana; and the MIS 5c Tewai Section indicator), AAR (MIS 5a Robe Range indicator), and other correlation methods (MIS 5a Atauro Island indicator). As above, we advocate revisiting sites with quality ratings of average (3) or poor (2) to better constrain the age uncertainty.”

We also added an additional paragraph to explain the WALIS standard for these ratings (revised manuscript lines 98–112 and lines 130 – 140):

“For each site we rated the quality of the RSL elevation data following criteria established by the World Atlas of Last Interglacial Shorelines project documentation (see Relative Sea Level at https://doi.org/10.5281/zenodo.3961544). Quality assessments for RSL elevation reflect a combination of measurement precision, the specificity of the reference datum for the elevation, and the range and uncertainty of the indicative meaning (*sensu* Rovere et al., 2016). When these three variables constrain total RSL uncertainty to <1 m or 1–2 m, then WALIS defines the RSL elevation as quality excellent (5) or good (4), respectively. If, however, uncertainties in these three variables lead to a RSL elevation of 2–3 m or > 3m, then a rating of average (3) or poor (2) applies, respectively. For sites without a specified reference datum (e.g., 3 m below sea level rather than 3 m below mean high tide), we limited the maximum quality rating to 3 and assigned a rating based on the remaining factors. Any RSL indicator of poorer quality than described above receives a very poor quality rating (1). Any terrestrial or marine limiting indicator that serves only as an upper or lower bound on RSL receives a rating of rejected (0). Not all primary

references report the indicative meaning of an RSL indicator, and thus for a subset of sites we calculated the indicative meaning with the IMCalc software (Lorscheid and Rovere, 2019) in order to assign an elevation quality rating.

For each site we rated the quality of the RSL chronology following criteria established by the World Atlas of Last Interglacial Shorelines project documentation (see Relative Sea Level at https://doi.org/10.5281/zenodo.3961544). Quality assessment of indicator age reflects how well the geochronology translates to a stage versus substage assignment. An excellent rating (5) attributes a RSL indicator to a narrow window within a substage of MIS 5 whereas a good rating (4) more generally assigns an RSL indicator to a substage. If geochronology only assigns an indicator to a generic interglacial (such as MIS 5), then this warrants an average rating (3). A poor rating (2) applies to incomplete chronologic data, or data that provides only a minimum or maximum age on the RSL indicator. Conflicting age assignments between marine isotope stages warrant a very poor quality rating (1). Finally, chronologic data unable to distinguish between two or more Pleistocene Epoch interglacials warrants a rejected quality rating (0)."

Finally, we added two Figures to illustrate the global distribution of the elevation and chronology quality ratings. These new Figures 6 and 7 can be found in the revised manuscript on pages 14 and 15.

**It would also be nice to have some brief commentary on why many of the sites are so important - many of the names resonate in the history of Quaternary Science and the understaing of Quaternary sea-level changes, and for that matter neotectonics.**

We restricted our commentary to the latter suggestion and have augmented manuscript Section 6, now titled "Review of Research Themes on MIS 5a and 5c RSL Indicators and Future Research Directions, and copy the new text below (revised manuscript line numbers 755–828).

"Two complementary research themes rely upon MIS 5 relative sea level indicators with unequivocal substage chronology and robust elevation data. One focus leverages the spatial variation in reconstructions of *local* MIS 5a and 5c RSL elevations to constrain global geophysical models for glacial isostatic adjustment and, hence, to refine estimates of substage global mean sea level (GMSL; e.g., Lambeck and Chappell, 2001; Potter and Lambeck, 2004; Muhs et al., 2012; Creveling et al., 2017). The other focus deduces regional rates of tectonic motion from the vertical displacement of (MIS 5e) RSL indicators from the sea level at which they formed (e.g., Matthews, 1973; Chappell, 1974; Wehmiller et al., 1977; Muhs et al., 1990, 1992b; Simms et al., 2016). Numerous field and numerical analyses highlight the entanglement of these themes. Robust efforts to deduce MIS 5a and 5c GMSL from misfit analyses between field observational data and GIA models necessitate a quantitative correction for a site's tectonic uplift history (Creveling et al., 2015; Simms et al., 2016). Inasmuch as a vertical tectonic uplift correction requires a robust paleo-sea level reference datum, this reference datum should reflect the estimates of interstadial GMSL *and* melt-induced spatial variations in local sea level from glacial isostatic adjustment models (Creveling et al., 2015; Simms et al., 2016). Advances in each research theme enrich the other, and both rely upon RSL indicators with high quality elevation measurements and substage-resolution chronology.

Tectonic uplift-corrected RSL indicators in the near-to-intermediate field of the North American ice complex display distinct geographic trends arising from glacial isostatic adjustment (Potter and Lambeck, 2004; Simms et al., 2016). North American Atlantic coast and Caribbean MIS 5a highstand elevations display a north-to-south latitudinal gradient that decreases by ~30 m elevation (Cronin et al., 1981, Szabo, 1985, Bard et al., 1990, Cutler et al., 2003, Potter et al., 2004, Wehmiller et al., 2004; Parham et al., 2013). Potter and Lambeck (2004) demonstrated that this trend reflects the glacio-isostatic disequilibrium imposed by the forebulge of the Laurentide ice sheet. After correcting Atlantic Coast and Caribbean RSL inferences for glacioisostasy, Potter and Lambeck (2004) concluded that MIS 5a GMSL peaked ~–28 m below present (with a similar value for MIS 5c). Potter and Lambeck (2004) predicted broadly consistent magnitudes, though narrower bounds, on MIS 5a and 5c substage GMSL as Lambeck and Chappell (2001) who reconstructed GMSL of 23–37 m and 18–30 m below present, respectively, from Huon Peninsula coral reef terraces. In contrast, tectonic uplift- and GIA-corrected MIS 5a and MIS 5c RSL indicators along the Pacific coast of the U.S. and Mexico reveal an opposing latitudinal gradient in local high-stand elevations from that observed on the North American Atlantic coast and Caribbean (Simms et al., 2016). On the basis of the North American Pacific coast geographic gradient, Simms et al. (2016) concluded that MIS 5 and 5c peak GMSL reached up to ~-15 m and ~-10 m below present sea level, a conclusion in agreement with that of Muhs et al. (2012) who reconstructed peak GMSL elevations of -16 m and -9 m during MIS 5a and MIS 5c, respectively, based on tectonic uplift- and GIA-corrected RSL indicators at San Nicolas Island, California; the Florida Keys; and Barbados.

The opposing latitudinal gradients in MIS 5a and 5c peak highstand elevations imposed by the peripheral bulge of the North American ice complex do not find reconciliation with conventional '1-D' glacial isostatic adjustment models that assume a depth-varying but laterally homogenous viscoelastic structure (Creveling et al., 2017). Notably, embedding an upper mantle viscosity in a GIA models to reconcile the highstand latitudinal gradient from one geographic region (i.e., the Pacific or Atlantic coast of North America) exacerbates the misfit of GIA predictions to the RSL indicators of the other region. Hence, GIA analyses that focus on a regional subset of global data produce GMSL estimates with systematic errors (hence the conflicting GMSL predictions of Potter and Lambeck (2004) versus Simms et al. (2016)). Creveling et al. (2017) promoted the adoption of a sensitivity analysis between globally distributed RSL indicators and GIA predictions that adopt viscosity models that honor the complexity in (North American) upper mantle viscosity. The resulting analytical workflow, applied to an unfiltered compendium of MIS 5a and 5c RSL indicators, yielded peak GMSL bounds of −18±1 m and −20±1 m for MIS 5a and MIS 5c, respectively; notably, repeating this sensitivity analysis on a RSL database filtered to include only those with high-quality (predominately uranium-series) chronology widened these bounds to −22±1 m and −24±2 m, respectively (Creveling et al., 2017).

Continued refinement of MIS 5a and 5c peak GMSL and regional rates of Quaternary vertical tectonic uplift remains within reach. First, numerical models for glacial isostatic adjustment that adopt '3D' solid earth models with depth-varying and laterally homogenous viscoelastic structure promise to reconcile observed spatial gradients in RSL highstands and GIA model predictions (e.g., Latychev et al., 2005; Clark et al., 2019). Such numerical advancements offer the possibility of refining GMSL estimates in the absence of further field data collection. Second, the quality

ratings conferred above motivate the strategic re-surveying of a subset of MIS 5a and 5c field observations (see Sections 5.1.19, 5.2.10, and 5.3.13) in order that each site conform to the uniform approach to establishing the elevation and uncertainty of elevation measurements ages adopted for the World Atlas of Last Interglacial Sea Level (e.g, Rovere et al., 2016). The re-sampling and/or re-analysis of geochronological material may also refine the numerical ages adopted for the World Atlas of Last Interglacial Sea Level (e.g, Rovere et al., 2016). Importantly, this retroactive translation of MIS 5a and 5c RSL observations to rigorous sea-level index points (*sensu* Hijma et al., 2015) offers the paired promise of refining predictions of contemporaneous global mean sea level and vertical tectonic motion and the standardization of efforts to complete these research foci. Third, the proliferation of airborne LiDAR data can offer geoscientists a fresh perspective on the quantity and spatial relationships of purported terrace platforms (Bowles and Cowgill, 2012) that, once ground-truthed, may confer confidence in, or contradict, chronologies developed from terrace counting methods. In practice, simultaneous efforts to enact all three practices will enrich conclusions about MIS 5a and 5c GMSL bounds and the accompanying tectonic displacement of these RSL indicators."

**Some references that have been overlooked could be included, such as;**

**Schellmann & Radtke (2004) Earth-Science Reviews, 64, 157-187 (for Barbados)**

**Blakemore et al. (2015) Marine Geology, 335, 377-383 (for Robe Range)**

**Schwebel, D. A. (1984) Quaternary stratigraphy and sea-level variation in the southeast of South Australia. In, B. G. Thom (Ed), Coastal Geomorphology in Australia (pp. 291-311), Academic Press, Sydney. (Outlines the stratigraphical nomenclature and numerical system for the ages of the interstadial barrier successions for Robe Range).**

**In a similar manner, and perhaps appallingly self-serving, the following reference may be of value about Robe Range in the context of the Coorong Coastal Plain;**

**Murray-Wallace, C. V. (2018) Quaternary history of the Coorong Coastal Plain, Southern Australia: An archive of environmental and global sea-level changes, Springer, Cham, 229 pp.**

> The bibliography and text now reference each of the above citations. Manuscript Figures 1–7 and Tables 1 and 2 now reflect these data additions.

**Some minor editorial comments:**

**Please avoid the term 'absolute' when applied in a geochronological context - nothing is absolute, apart from death and taxes. I would suggest the term 'numeric'. Although 'aged' the following reference is of value in this regard;**

**Colman et al. (1987) Suggested terminology for Quaternary dating methods. Quaternary Research, 28, 314-319.**

We have adopted the term 'numeric' to replace 'absolute' throughout the text.

We have implemented all of the referee's line edits and terminology suggestions detailed below. We have also carefully reviewed the text for grammatical errors and fixed any found.

**I would suggest in the title and all subsequent instances using the expression Sub-stage 5a abnd 5c**

**Line 8 document instead of detail**

**Line 17 Eartth's**

**Line 20 and all subsequent instances 'highstand' or 'highstands' (as recognised by Sequence Stratigraphy)**

**Figure 4 caption, line 2; 'geochronological'**

**'Ages' instead of 'dates', the latter being unique calendar events or 'hot nights out'**

**I wondered about the terminology of 'wave-cut platforms' as there has been much controversy on the use of this term in view of the processes that shape platforms - perhaps use the non-genetic term 'shore platform'**

**Line 142 with the Whiskey (on that matter - correct spelling of Whisky or Whiskey? and in subsequent instances)**

**Line 166 with the MIS 5c Pioneer**

**Line 271 perhaps changed 'posited' to 'argued' or 'suggested'**

**It might be worth having some commentary on the reliability of the U-series ages in the context of the reported ages and delta234U values, where this is possible.**

**Line 315 delete second instance of 'overall' - on the same line, I am not sure what 'have very good chronologies' means? In what sense? Please clarify.**

**Line 429 species in italics**

**In terms of the Huon Peninsula, I personally feel that it is critical to include the 'Reconciliation' paper by Chappell et al. (1996) Earth and Planetary Science Letters, 141, 227-236.**

**Line 445 delete second instance of reference**

Line 480 luminescene has more commonly been used in these regions as the method determines the timing (age) of the depositional event, and also because of the paucity of corals in these successions, that would otherwise have been appropriate for U-series dating.

Line 505 word choice in terms of 'densely' - is this really true?

I don't know if it is possible, however, some photographs of some classic field sites would aid the visual appeal of the paper.

Colin Murray-Wallace

11th February 2021

**Referee #2 (Anonymous)**

**General comments:**

**In this review Thomspon and Creveling compiled the marine terraces and paleoshoreline sea-level indicators that formed during the interstadials MIS 5a and 5c. The authors divide the geographical distribution of the indicators in 3 main regions: Pacific coast of North America, the Atlantic coast of North America and the Caribbean, and the remaining globe. This global compilation includes the elevation, indicative meaning, and chronology of the indicators. Due to its global context, this component of the WALIS database will prove to be very useful by facilitating global sea level reconstructions and contributing to refining the corrections needed for glacial isostatic adjustments and regional tectonic deformation.**

We thank the anonymous referee for their positive summary of our manuscript.

**I think the manuscript is overall well written and concise. The majority of the manuscript deals with reporting the measured elevations and chronologies of the MIS 5 and 5c indicators, however, in my opinion, this work would benefit from a discussion before Future research directions on the GIA effects and tectonic deformation. I suggest the authors to address how these indicators are useful to facilitate the future investigations of GIA models, particularly given the good coverage in the near field of the North American Ice Sheets, as well as their usefulness for better constraints of the Quaternary tectonic deformation.**

This comment echoes one from Dr. Murray-Wallace's review, and therefore we focused much effort on addressing this recommendation. The resulting text additions to the manuscript can be found under the response to Dr. Murray-Wallace (revised manuscript line numbers 755 – 828). We followed the referee's content and organization suggestion to craft this new section.

**Although there is a brief summary section at the end of each of the 3 different regions, it seems that a section of more general conclusions of this compilation in the end of the manuscript is missing.**

To address the reviewer's comment, we chose to elaborate on our regional summary sections (see revised manuscript lines 395 – 447, 548 – 580, and 684 – 718).

**Specific comments:**

**I suggest using "substages" in the title and throughout the text.**

We now adopt 'substage' throughout the title and text.

**Please check the references throughout the manuscript and consistently use "et al" with non-italic, as per journal guidelines.**

We have corrected this formatting.

**The authors use "uranium-thorium", "uranium-series", "uranium series" dating - Please choose one of these and use it consistently throughout the text.**

We now apply the phrase "uranium-series" throughout the text.

**I suggest including a brief discussion about the age quality and indicator quality presented in Tables 1 and 2 and refer the reader to the evaluation guide by which indicators are rated on a 0 (rejected) to 5 (excellent) scale.**

As documented in response to Dr. Murray-Wallace's review, we now include comprehensive discussion of quality estimates (revised manuscript lines 395 – 447, 548 – 580, and 684 – 718).

We implemented every line edit request listed below.

**Details:**

**Line 7: I suggest delete "and detail".**

**Line 17: Earth's**

**Line 39: delete "with MIS 5a and 5c paleo-sea level indicators" - it's already mentioned at the beginning of the sentence**

**Line 39: "includes sites". Isn't 39 the number of total sites, instead of 36?**

**Line 42: uncertainty - do authors refer to Elevation measurements' uncertainty here? Not clear.**

**Line 51: reflects**

**Line 52: introduce here the acronym GIA**

**Line 60: delete GIA**

**Line 61: delete GMSL - has already been mentioned in line 53**

**Line 69: Muhs et al 1992b? But Muhs 1992a hasn't been cited yet**

**Line 73: eolianites**

**Line 89: introduce the acronym AAR**

**Line 96: radiocarbon dating**

Line 105: "present review"?

Line 135: delete "amino acid racemization" and keep only AAR

Line 150: see "above" not below

Line 155: this is the same sentence as in lines 146-147

Line 157: mention the age?

Line 177: add "respectively"

Line 178: I suggest rephrasing this sentence: "Merritts and Bull (1989) assigned the 10 m apsl and 23 m apsl terraces to MIS 5a and 5c" has been mentioned already 2 lines above

Line 221: bones?

Line 222: "." missing after (figure 4).  "Corals" instead of "coral".

Line 225: I suggest avoiding to use the word "terrace" so many times (i.e., 3 times in one sentence).

Line 214: delete "." after fossiliferous

Line 245: I suggest using "open-system behavior"

Line 251: delete comma

Line 252: delete first comma

Line 259: delete comma

Line 287: delete apsl

Line 309: America

Line 312: the cited reference is missing the year

Line 315: delete the second "overall"

Line 321: "." missing at the end of the sentence

Line 323: corals

Line 325: is it age assignment of MIS 5a instead of MIS 5 here?

**Line 348: I suggest deleting "aged"**

**Line 351: "represents". I am not clear what the authors mean by " specific Fig. 5"?**

**Line 364: "radiocarbon ages and uranium-series dates" - to avoid confusion, I recommend to clarify the difference between "a date" and "an age" and use it correspondingly throughout the manuscript.**

**Line 368: eolianites**

**Line 377: fits**

**Line 390: corals**

**Line 391: use comma before "respectively"**

**Line 410: delete "terrace" before lowest. "is mapped"**

**Line 419: delete "OSL"**

**Line 428-429: use italic for coral species**

**Line 439: corals**

We implemented every Figure and Figure caption request listed below. We also made minor corrections to the Figures.

**Figure 1: I would suggest a clear separation between panel (a-c) from (d-f) (i.e., move (d-f) more to the right, otherwise I find reading the figure a bit confusing).**

**Figure 1, 2, 3, 4, and 5 captions: use RSL and delete "relative sea level"**

**Figure 4 and 5: I recommend placing the legend in a better position so that it doesn't overlap with the data (one suggestion would be to have dates and substage assignment on two different columns).**

In addition to the referee-requested edits detailed above, we received two unsolicited emails related to the WALIS manuscript from Drs. Wehmiller and Bard. To address the former, we now include in the WALIS database, and in the manuscript text, reference to two additional uranium-series dates for Pamlico Sound, as well as a reference to a regional database with numerous MIS 5 to Holocene AAR ages (revised manuscript lines 467 – 470). For the latter, we added additional uranium-series ages for the Barbados sites to the WALIS database, and the appropriate reference to the bibliography. Based on advice from Dr. Rovere, we now cite Chutcharavan and Dutton (2021) for their uranium-series entries relevant to MIS 5a and 5c. Finally, based on our careful reading of the text and tables, we fixed an error related to the Santa Cruz Highway 1 elevation (now reads 26 – 39 m) and we adopted consistent

terminology for the North American Pacific and Atlantic coasts by removing reference to the U.S. west and east coast, respectively.

We thank you and the referees for the opportunity to strengthen our manuscript.

Warm regards,
Schmitty Thompson and Jessica Creveling

---

## Author Response (AR2)

Dear Dr. Deirdre Ryan,

Thank you for your insightful comments on our revised manuscript. Enclosed you will find a revised manuscript text and figures that incorporate your requested corrections.

(1) For those sites with chronologies based solely on amino acid racemization, we updated to quality ratings to better reflect the precision of the age assignments. This upwardly revised the MIS 5a and 5c Gaviota terraces and MIS 5a Newport, Santa Rosa Island, Oceanside, and Eleuthera Island indicators age quality ratings from 2 to 4 and raised the MIS 5a Bruhel Point and Robe Range indicators age quality rating from 2 to 3.

The corresponding summary sections (5.1.19, 5.2.10, 5.3.13) figures (6 and 7) and tables (1 and 2) now reflect these updated quality ratings.

(2) We also revised the terminology of "wave-cut platform" to "shore platform" in sections 1, 2, 5.1.2, 5.1.4, 5.1.8, 5.1.12, and 5.3.3.

(3) We have made strides in converting Figures 4, 5, 6 and 7 into a more color-blind friendly color palette, and we attach those to this revision should it be possible to include these within the revised manuscript.

We are also going to upload a version 2 of our WALIS data into the Zenodo data repository to incorporate changes made to the database during the revision process. At what stage will we be able to update the link in the manuscript?

We will endeavor to take into account your suggestions for future manuscripts.

Sincerely,

-Schmitty Thompson and Jessica Creveling